# Model Selection for Bayesian Autoencoders

**Ba-Hien Tran**
EURECOM
(France)

**Simone Rossi**
EURECOM
(France)

**Dimitrios Milios**
EURECOM
(France)

**Pietro Michiardi**
EURECOM
(France)

**Edwin V. Bonilla**
CSIRO's Data61
The Australian National University
The University of Sydney
(Australia)

**Maurizio Filippone**
EURECOM
(France)

## Abstract

We develop a novel method for carrying out model selection for Bayesian autoencoders (BAEs) by means of prior hyper-parameter optimization. Inspired by the common practice of type-II maximum likelihood optimization and its equivalence to Kullback-Leibler divergence minimization, we propose to optimize the distributional sliced-Wasserstein distance (DSWD) between the output of the autoencoder and the empirical data distribution. The advantages of this formulation are that we can estimate the DSWD based on samples and handle high-dimensional problems. We carry out posterior estimation of the BAE parameters via stochastic gradient Hamiltonian Monte Carlo and turn our BAE into a generative model by fitting a flexible Dirichlet mixture model in the latent space. Consequently, we obtain a powerful alternative to variational autoencoders, which are the preferred choice in modern applications of autoencoders for representation learning with uncertainty. We evaluate our approach qualitatively and quantitatively using a vast experimental campaign on a number of unsupervised learning tasks and show that, in small-data regimes where priors matter, our approach provides state-of-the-art results, outperforming multiple competitive baselines.

## 1 Introduction

The problem of learning useful representations of data that facilitate the solution of downstream tasks such as clustering, generative modeling and classification, is at the crux of the success of many machine learning applications [see, e.g., 5, and references therein]. From a plethora of potential solutions to this problem, unsupervised approaches based on autoencoders [10] are particularly appealing as, by definition, they do not require label information and have proved effective in tasks such as dimensionality reduction and information retrieval [20].

Autoencoders are neural network models composed of two parts, usually referred to as the encoder and the decoder. The encoder maps input data to a set of lower-dimensional latent variables. The decoder maps the latent variables back to the observations. The bottleneck introduced by the low-dimensional latent space is what characterizes the compression and representation learning capabilities of autoencoders. It is not surprising that these models have connections with principal component analysis [4], factor analysis and density networks [33], and latent variable models [28].

In applications where quantification of uncertainty is a primary requirement or where data is scarce, it is important to carry out a Bayesian treatment of these models by specifying a prior distribution over their parameters, i.e., the weights of the encoder/decoder. However, estimating the posterior

35th Conference on Neural Information Processing Systems (NeurIPS 2021).

distribution over the parameters of these models, which we refer to as Bayesian autoencoders (BAEs), is generally intractable and requires approximations. Furthermore, the need to specify priors for a large number of parameters, coupled with the fact that autoencoders are not generative models, has motivated the development of Variational Autoencoders (VAEs) as an alternative that can overcome these limitations [25]. Indeed, VAEs have found tremendous success and have become one of the preferred methods in modern machine-learning applications [see, e.g., 26, and references therein].

To recap, three potential limitations of BAEs hinder their widespread applicability in order to achieve a similar or superior adoption to their variational counterpart: (i) lack of generative modeling capabilities; (ii) intractability of inference and (iii) difficulty of setting sensible priors over their parameters. In this work we revisit BAEs and deal with these limitations in a principled way. In particular, we address the first limitation in (i) by employing density estimation in the latent space. Furthermore, we deal with the second limitation in (ii) by exploiting recent advances in Markov chain Monte Carlo (MCMC) and, in particular, stochastic gradient Hamiltonian Monte Carlo (SGHMC) [9]. Finally, we believe that the third limitation (iii), which we refer to as the difficulty of carrying out *model selection*, requires a more detailed treatment because choosing sensible priors for Bayesian neural networks is an extremely difficult problem, and this is the main focus of this work.

**Contributions.** Specifically, in this paper we provide a novel, practical, and elegant way of performing model selection for BAEs, which allows us to revisit these models for applications where VAEs are currently the primary choice. We start by considering the common practice of estimating prior (hyper-)parameters via type-II maximum likelihood, which is equivalent to minimizing the Kullback-Leibler divergence (KL) between the distribution induced by the BAE and the data generating distribution. Because of the intractability of this objective and the difficulty to estimate it through samples, we resort to an alternative formulation where we replace the KL with the distributional sliced-Wasserstein distance (DSWD) between these two distributions. The advantages of this formulation are that we can estimate the DSWD based on samples and, thanks to the slicing, we can handle large dimensional problems. Once BAE hyper-parameters are optimized, we estimate the posterior distribution over the BAE parameters via SGHMC [9], which is a powerful sampler that operates on mini-batches and has proven effective for Bayesian deep/convolutional networks [45, 52, 23]. Furthermore, we turn our BAE into a generative model by fitting a flexible mixture model in the latent space, namely the Dirichlet Process Mixture Model (DPMM). We evaluate our approach qualitatively and quantitatively using a vast experimental campaign on a number of unsupervised learning tasks, with particular emphasis on the challenging task of generative modeling when the number of observations is small.

## 2 Preliminaries on Bayesian Autoencoders

An autoencoder (AE) is a neural network parameterized by a set of parameters $\mathbf{w}$, which transforms an unlabelled dataset, $\mathbf{x} \stackrel{\text{def}}{=} \{\mathbf{x}_n\}_{n=1}^N$, into a set of reconstructions $\mathbf{f} \stackrel{\text{def}}{=} \{\mathbf{f}_n\}_{n=1}^N$, with $\mathbf{x}_n, \mathbf{f}_n \in \mathbb{R}^D$. An AE is composed of two parts: (1) an encoder $f_{\text{enc}}$ which maps an input sample $\mathbf{x}_n$ to a latent code $\mathbf{z}_n \in \mathbb{R}^K$, $K \ll D$; and (2) a decoder $f_{\text{dec}}$ which maps the latent code to a reconstructed datapoint $\mathbf{f}_n$. In short, $\mathbf{f} = f(\mathbf{x}; \mathbf{w}) = (f_{\text{dec}} \circ f_{\text{enc}})(\mathbf{x})$, where we denote $\mathbf{w} := \{\mathbf{w}_{\text{enc}}, \mathbf{w}_{\text{dec}}\}$ the union of parameters of the encoder and decoder.

The Bayesian treatment of AEs dictates that a prior distribution $p(\mathbf{w})$ is placed over all parameters of $f_{\text{enc}}$ and $f_{\text{dec}}$ and the posterior is inferred using Bayes' rule. For the sake of presentation, we can consider a BAE as a classic Bayesian neural network (BNN) for a supervised learning task [34, 37], where we have labels $\mathbf{y} \stackrel{\text{def}}{=} \{\mathbf{y}_n\}_{n=1}^N$ associated with each input point $\mathbf{x}_n$. From this perspective, we can write the posterior on $\mathbf{w}$ as follows:

$$p(\mathbf{w} \mid \mathbf{y}, \mathbf{x}) = \frac{p(\mathbf{y} \mid \mathbf{w}, \mathbf{x})p(\mathbf{w})}{p(\mathbf{y} \mid \mathbf{x})}, \tag{1}$$

where $p(\mathbf{y} \mid \mathbf{w}, \mathbf{x})$ is the likelihood defined by the network architecture and the denominator—$p(\mathbf{y} \mid \mathbf{x})$—constitutes the marginal likelihood. In order to keep the notation uncluttered, we can drop the explicit dependency on $\mathbf{x}$ as input, which leads to:

$$p(\mathbf{w} \mid \mathbf{y}) = \frac{p(\mathbf{y} \mid \mathbf{w})p(\mathbf{w})}{p(\mathbf{y})}, \tag{2}$$

We observe that for an AE the labels $\mathbf{y}$ are the same as the datapoints $\mathbf{x}$, meaning that the likelihood is computed in $\mathbf{x}$.

**Likelihood model.** Before giving an in-depth treatment on priors for BAEs in the next section, we briefly discuss the likelihood, which can be chosen according to the type of data. Firstly, we assume factorization of the likelihood on the datapoints, i.e. $p(\mathbf{y} \mid \mathbf{w}) = \prod_{n=1}^{N} p(\mathbf{y}_n \mid \mathbf{w})$. Secondly, given that our experiments mainly focuses on image datasets where pixel values are normalized in the $[0, 1]$ range, we will use the *continuous Bernoulli* distribution [31]:

$$p(\mathbf{y}_n \mid \mathbf{w}) = \prod_{i=1}^{D} K(\lambda_i) \lambda_i^{\mathbf{y}_{n,i}} (1 - \lambda_i)^{1 - \mathbf{y}_{n,i}} \coloneqq p(\mathbf{y}_n \mid \mathbf{f}_n), \tag{3}$$

where $K(\lambda_i)$ is a properly defined normalization constant [31] and $\lambda_i = f_i(\mathbf{x}_n; \mathbf{w}) = \mathbf{f}_{n,i} \in [0, 1]$ is the $i$-th output from the BAE given the input $\mathbf{x}_n$. We note that, as $\mathbf{f}_n$ depends deterministically on $\mathbf{w}$, we will use the above expression to refer to both $p(\mathbf{y}_n \mid \mathbf{w})$ and $p(\mathbf{y}_n \mid \mathbf{f}_n)$, where the latter term will be of crucial importance when we define the functional prior induced over the reconstruction $\mathbf{f}$. Finally, we remark that in the Bayesian scheme both the prior and likelihood are modeling choices. In fact, in our experiments we explore an additional likelihood model, namely the *truncated Gaussian*, and we show how the problem of selecting good priors is orthogonal to the choice of the likelihood.

**Inference.** Although the posterior over the BAE parameters is analytically intractable, it can be approximated by variational methods or using MCMC sampling. Within the large family of approximate Bayesian inference schemes, SGHMC [9] allows us to sample from the true posterior by efficiently simulating a Hamiltonian system [38]. Unlike more traditional methods, SGHMC can scale up to large datasets by relying on noisy but unbiased estimates of the potential energy function $U(\mathbf{w}) = -\sum_{n=1}^{N} \log p(\mathbf{y}_n \mid \mathbf{w}) - \log p(\mathbf{w})$. These can be computed by considering a mini-batch of size $M$ of the data and approximating $\sum_{n=1}^{N} \log p(\mathbf{y}_n \mid \mathbf{w}) \approx \frac{N}{M} \sum_{j \in \mathcal{I}_M} \log p(\mathbf{y}_j \mid \mathbf{w})$, where $\mathcal{I}_M$ is a set of $M$ random indices. More details on SGHMC can be found in the Appendix.

**Pathologies of standard priors.** The choice of the prior is important for the Bayesian treatment of any model as it characterizes the hypothesis space [34, 36]. Specifically for BAEs, one should note that placing a prior on the parameters of the encoder and decoder has an implicit effect on the prior over the network output (i.e. the reconstruction). In addition, the highly nonlinear nature of these models implies that interpreting the effect of the architecture is theoretically intractable and practically challenging. Several works argue that a vague prior such as $\mathcal{N}(0, 1)$ is good enough for some tasks and models, like classification with convolutional neural networks (CNNs) [49].

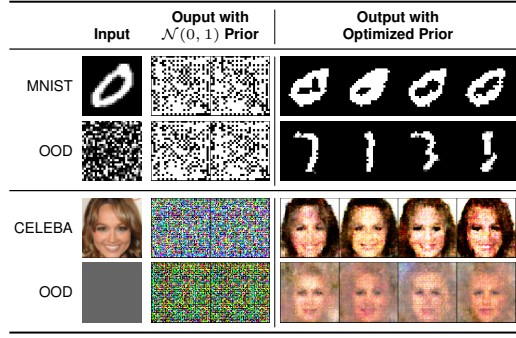

**Figure 1:** Realizations sampled from different priors given an input image. OOD stands for out-of-distribution.

However, for BAEs this is not enough, as illustrated in Fig. 1. The realizations obtained by sampling weights/biases from a $\mathcal{N}(0, 1)$ prior indicate that this choice provides poor inductive bias. Meanwhile, by encoding better beliefs via an optimized prior, which is the focus of the next section, the samples can capture main characteristics intrinsic to the data, even when the model is fed with out-of-distribution inputs.

## 3 Model Selection for Bayesian Autoencoders via Prior Optimization

One of the main advantages of the Bayesian paradigm is that we can incorporate prior knowledge into the model in a principled way. Let us assume a prior distribution $p_\psi(\mathbf{w})$ on the parameters of the AE network, where now we are explicit on the set of (hyper-)parameters that determine the prior, i.e., $\psi$. Specifying the prior is easy, e.g., a Gaussian. Determining the effective functional prior, i.e., the prior over the network output $\mathbf{f}$ is not trivial due to the complex nonlinear forms of $f_{\text{enc}}$ and $f_{\text{dec}}$,

which induce a non-trivial effect on the output (functional) prior:

$$p_{\boldsymbol{\psi}}(\mathbf{f}) = \int f(\mathbf{x}; \mathbf{w}) p_{\boldsymbol{\psi}}(\mathbf{w}) \mathrm{d}\mathbf{w}, \tag{4}$$

where, as before, $\mathbf{f} = f(\mathbf{x}; \mathbf{w})$ is the functional output of the BAE. Although $p_{\boldsymbol{\psi}}(\mathbf{f})$ cannot be evaluated analytically, it is possible to draw samples from it.

**Prior parameterization.** The only two requirements needed to design a parameterization for the prior are: to be able to (1) draw samples from it and (2) to compute its log-density at any point. The latter is required by many inference algorithms such as SGHMC. We consider a fully-factorized Gaussian prior over weights and biases at layer $l$:

$$p(w_l) = \mathcal{N}(w_l; \mu_{l_w}, \sigma_{l_w}^2), \quad p(b_l) = \mathcal{N}(b_l; \mu_{l_b}, \sigma_{l_b}^2), \tag{5}$$

Notice that, as we shall see in § 3.2 and § 3.3, in order to estimate our prior hyper-parameters, we will require gradient back-propagation through the stochastic variables $w_l$ and $b_l$. Thus, we treat these parameters in a deterministic manner by means of the reparameterization trick [42, 25].

## 3.1 Another route for Bayesian Occam's razor

A common way to estimate hyper-parameters (i.e., prior parameters $\boldsymbol{\psi}$) is to rely on the Bayesian Occam's razor (a.k.a. *empirical Bayes*), which dictates that the marginal likelihood $p_{\boldsymbol{\psi}}(\mathbf{y})$ should be optimized with respect to $\boldsymbol{\psi}$. There are countless examples where such simple procedure succeeds in practice [see, e.g., 41, 21]. We note however that marginal likelihood maximization for a large number of hyper-parameters can suffer from overfitting [41, 40]. Nevertheless, we do not expect significant overfitting issues in our setting, as we focus on data that are characterized by a high level of structure (i.e. images). As we have seen, regular choices for the prior completely fail to capture the properties of such highly-structured outputs.

The marginal likelihood is obtained by marginalizing out the outputs $\mathbf{f}$ and the model parameters $\mathbf{w}$,

$$p_{\boldsymbol{\psi}}(\mathbf{y}) = \int p(\mathbf{y} \,|\, \mathbf{f}) p_{\boldsymbol{\psi}}(\mathbf{f}) \mathrm{d}\mathbf{f}, \tag{6}$$

where $p(\mathbf{y} \,|\, \mathbf{f})$ and $p_{\boldsymbol{\psi}}(\mathbf{f})$ are given by Eq. 3 and Eq. 4, respectively. Unfortunately, in our context it is impossible to carry out this optimization due to the intractability of Eq. 6.

Classic results in the statistics literature draw parallels between maximum likelihood estimation (MLE) and KL minimization [2],

$$\arg\max_{\boldsymbol{\psi}} \int \pi(\mathbf{y}) \log p_{\boldsymbol{\psi}}(\mathbf{y}) \mathrm{d}\mathbf{y} = \arg\min_{\boldsymbol{\psi}} \underbrace{\int \pi(\mathbf{y}) \log \frac{\pi(\mathbf{y})}{p_{\boldsymbol{\psi}}(\mathbf{y})} \mathrm{d}\mathbf{y}}_{\text{KL}[\pi(\mathbf{y}) \,\|\, p_{\boldsymbol{\psi}}(\mathbf{y})]}, \tag{7}$$

where $\pi(\mathbf{y})$ is the true data distribution. This equivalence provides us with an interesting insight on an alternative view of marginal likelihood optimization as minimization of the divergence between the true data distribution and the marginal $p_{\boldsymbol{\psi}}(\mathbf{y})$.

This alternative view allows one to obtain a viable optimization strategy that relies on an empirical estimate of the data distribution $\tilde{\pi}(\mathbf{y})$. This presents additional challenges however, as the empirical evaluation and optimization of KL divergences remains a well-known challenging problem [13]. Although it is possible to evaluate KL (or any other $f$-divergence) empirically by leveraging results from convex analysis [39], we have opted to substitute KL with an alternative metric that is more convenient from a computational perspective. We are inspired by recent works on generative adversarial networks [3, 16] and Bayesian neural networks [45], where it is shown that the Wasserstein distance can be estimated efficiently using samples only, even for high-dimensional distributions. We thus employ the Wasserstein distance as a surrogate for KL divergence, so that we avoid the challenges of empirical KL estimation.

To summarize: (1) we would like to do prior selection by carrying out type-II MLE; (2) the MLE objective is analytically intractable but the connection with KL minimization allows us to (3) swap the divergence with the Wasserstein distance, yielding a practical framework for choosing priors.

## 3.2 Matching the marginal distribution to the data distribution via Wasserstein distance minimization

Given the two probability measures $\pi$ and $p_{\boldsymbol{\psi}}$, both defined on $\mathbb{R}^D$ for simplicity, the $p$-Wasserstein distance between $\pi$ and $p_{\boldsymbol{\psi}}$ is given by

$$W_p^p(\pi, p_{\boldsymbol{\psi}}) = \inf_{\gamma \in \Gamma(\pi, p_{\boldsymbol{\psi}})} \int \|\mathbf{y} - \mathbf{y}'\|^p \gamma(\mathbf{y}, \mathbf{y}') \mathrm{d}\mathbf{y}\mathrm{d}\mathbf{y}', \tag{8}$$

where $\Gamma(\pi, p_{\boldsymbol{\psi}})$ is the set of all possible distributions $\gamma(\mathbf{y}, \mathbf{y}')$ such that the marginals are $\pi(\mathbf{y})$ and $p_{\boldsymbol{\psi}}(\mathbf{y}')$ [47]. While usually analytically unavailable or computationally intractable, for $D = 1$ the distance has a simple closed form solution, that can be easily estimated using samples only [27].

The distributional sliced-Wasserstein distance (DSWD) takes advantage of this result by projecting the estimation of distances for high-dimensional distributions into simpler estimation of multiple distances in one dimension. The projection is done using the Radon transform $\mathcal{R}$, an operator that maps a generic density function $\varphi$ defined in $\mathbb{R}^D$ to the set of its integrals over hyperplanes in $\mathbb{R}^D$,

$$\mathcal{R}\varphi(t, \boldsymbol{\theta}) := \int \varphi(\mathbf{r})\delta(t - \mathbf{r}^\top \boldsymbol{\theta})\mathrm{d}\mathbf{r}, \quad \forall t \in \mathbb{R}, \ \forall \boldsymbol{\theta} \in \mathbb{S}^{D-1}, \tag{9}$$

where $\mathbb{S}^{D-1}$ is the unit sphere in $\mathbb{R}^D$ and $\delta(\cdot)$ is the Dirac delta [17]. Using the Radon transform, for a given direction (or *slice*) $\boldsymbol{\theta}$ we can project the two densities $\pi$ and $p_{\boldsymbol{\psi}}$ into one dimension and we can solve the optimal transport problem in this projected space. Furthermore, to avoid unnecessary computations, instead of considering all possible directions in $\mathbb{S}^{D-1}$, DSWD proposes to find the optimal probability measure of slices $\sigma(\boldsymbol{\theta})$ on the unit sphere $\mathbb{S}^{D-1}$,

$$DSW_p(\pi, p_{\boldsymbol{\psi}}) := \sup_{\sigma \in \mathbb{M}_C} \left( \mathbb{E}_{\sigma(\boldsymbol{\theta})} W_p^p \big( \mathcal{R}\pi(t, \boldsymbol{\theta}), \mathcal{R}p_{\boldsymbol{\psi}}(t, \boldsymbol{\theta}) \big) \right)^{1/p}, \tag{10}$$

where, for $C > 0$, $\mathbb{M}_C$ is the set of probability measures $\sigma$ such that $\mathbb{E}_{\boldsymbol{\theta}, \boldsymbol{\theta}' \sim \sigma}[\boldsymbol{\theta}^\top \boldsymbol{\theta}'] \leq C$ (a constraint that aims to avoid directions to lie in only one small area). The direct computation of $DSW_p$ in Eq. 10 is still challenging but admits an equivalent dual form,

$$\sup_{h \in \mathcal{H}} \left\{ \left( \mathbb{E}_{\bar{\sigma}(\boldsymbol{\theta})} \big[ W_p^p \big( \mathcal{R}\pi(t, h(\boldsymbol{\theta})), \mathcal{R}p_{\boldsymbol{\psi}}(t, h(\boldsymbol{\theta})) \big) \big] \right)^{1/p} - \lambda_C \mathbb{E}_{\boldsymbol{\theta}, \boldsymbol{\theta}' \sim \bar{\sigma}} \big[ |h(\boldsymbol{\theta})^\top h(\boldsymbol{\theta}')| \big] \right\} + \lambda_C C, \tag{11}$$

where $\bar{\sigma}$ is a uniform distribution in $\mathbb{S}^{D-1}$, $\mathcal{H}$ is the set of functions $h : \mathbb{S}^{D-1} \to \mathbb{S}^{D-1}$ and $\lambda_C$ is a regularization hyper-parameter. The formulation in Eq. 11 is obtained by employing the Lagrangian duality theorem and by reparameterizing $\sigma(\boldsymbol{\theta})$ as push-forward transformation of a uniform measure in $\mathbb{S}^{D-1}$ via $h$. Now, by parameterizing $h$ using a deep neural network with parameters $\boldsymbol{\phi}$, defined as $h_{\boldsymbol{\phi}}$, Eq. 11 becomes an optimization problem with respect to the network parameters. The final step is to approximate the analytically intractable expectations with Monte Carlo integration,

$$\max_{\boldsymbol{\phi}} \left\{ \left[ \frac{1}{K} \sum_{i=1}^K \big[ W_p^p \big( \mathcal{R}\pi(t, h_{\boldsymbol{\phi}}(\boldsymbol{\theta}_i)), \mathcal{R}p_{\boldsymbol{\psi}}(t, h_{\boldsymbol{\phi}}(\boldsymbol{\theta}_i)) \big) \big] \right]^{1/p} - \frac{\lambda_C}{K^2} \sum_{i,j=1}^K |h_{\boldsymbol{\phi}}(\boldsymbol{\theta}_i)^\top h_{\boldsymbol{\phi}}(\boldsymbol{\theta}_j)| \right\} + \lambda_C C,$$

with $\boldsymbol{\theta}_i \sim \bar{\sigma}(\boldsymbol{\theta})$. Finally, we can use stochastic gradient methods to update $\boldsymbol{\phi}$ and then use the resulting optima for the estimation of the original distance. We encourage the reader to check the detailed explanation of this formulation, including its derivation and some practical considerations for implementation, available in the Appendix.

## 3.3 Summary

We aim at learning the prior on the BAE parameters by optimizing the marginal $p_{\boldsymbol{\psi}}(\mathbf{y})$ obtained after integrating out the weights from the joint $p_{\boldsymbol{\psi}}(\mathbf{y}, \mathbf{w})$. The connection with *empirical Bayes* and KL minimization suggests that we can find the optimal $\boldsymbol{\psi}^\star$ by minimizing the KL between the true data distribution $\pi(\mathbf{y})$ and the marginal $p_{\boldsymbol{\psi}}(\mathbf{y})$. However, matching these two distributions is non-trivial due to their high dimensionality and the unavailability of their densities. To overcome this problem, we propose a sample-based approach using the distributional sliced 2-Wasserstein distance (Eq. 11) as objective:

$$\boldsymbol{\psi}^\star = \arg\min_{\boldsymbol{\psi}} \left[ DSW_2 \big( p_{\boldsymbol{\psi}}(\mathbf{y}), \pi(\mathbf{y}) \big) \right]. \tag{12}$$

This objective function is flexible and does not require the closed-form of either $p_\psi(\mathbf{y})$ or $\pi(\mathbf{y})$. The only requirement is that we can draw samples from these two distributions. Note that we can sample from $p_\psi(\mathbf{y})$, by first computing $\mathbf{f}$ after sampling from $p_\psi(\mathbf{w})$ and then perturbing the generated $\mathbf{f}$ by sampling from the likelihood $p(\mathbf{y} \mid \mathbf{f})$. For the continuous Bernoulli likelihood this operation can be implemented by using the reparameterization form that allows to backpropagate gradients [31].

## 4    Experiments

**Competing approaches.** We compare our proposal with a wide selection of methods from the literature. For autoencoding methods, we choose the vanilla **VAE** [25], the $\boldsymbol{\beta}$-**VAE** [19] and **WAE** (Wasserstein AE) [43]. In addition, we consider models with more complex encoders (**VAE + Sylvester flows** [46]), generators (**2-stage VAE** [11]), and priors (**VAE + VampPrior** [44]). For CELEBA we also include a comparison with Generative Adversarial Networks (GANs), with the vanilla setup of **NS-GAN** [15, 32] and the more recent **DiffAugment-GAN** [53, 24]. Finally, we also compare against BAE with the standard $\mathcal{N}(0, 1)$ prior. Unless otherwise stated, all models—including ours—share the same latent dimensionality ($K = 50$). We defer a more detailed description of these models and architectures to the Appendix.

**Generative process.** Differently from VAEs and other methods, deterministic and Bayesian AEs are not generative models. To generate new samples with BAEs we employ ex-post density estimation over the learned latent space, by fitting a density estimator $p_\vartheta(\mathbf{z})$ to $\{\mathbf{z}_i = \mathbb{E}_{p(\mathbf{w}_{\text{enc}} \mid \mathbf{y})}[f_{\text{enc}}(\mathbf{x}_i; \mathbf{w}_{\text{enc}})]\}$. In this work, we employ a nonparametric model for density estimation based on Dirichlet Process Mixture Model (DPMM) [7], so that its complexity is automatically adapted to the data; see also [6] for alternative ways to turn AEs into generative models. After estimating $p_\vartheta(\mathbf{z})$, a new sample can be generated by drawing $\mathbf{z}_{\text{new}}$ from $p_\vartheta(\mathbf{z})$ and $\mathbf{f}_{\text{new}} = \mathbb{E}_{p(\mathbf{w}_{\text{dec}} \mid \mathbf{y})}[f_{\text{dec}}(\mathbf{z}_{\text{new}}; \mathbf{w}_{\text{dec}})]$.

**Evaluation metrics.** To evaluate the reconstruction quality, we use the test log-likelihood (LL), which tells us how likely the test targets are generated by the corresponding model. The predictive log-likelihood is a proper scoring rule that depends on both the accuracy of predictions and their uncertainty [14]. To assess the quality of the generated images, instead, we employ the widely used Fréchet Inception Distance (FID) [18]. We note that, as GANs are not inherently equipped with an explicit likelihood model, we only report their FID scores. Finally, all our experiments and evaluations are repeated four times, with different random training splits.

### 4.1    Analysis of the effect of the prior

To demonstrate the effect of our model selection strategy, we consider scenarios in the small-data regime where the prior might not be necessarily tuned on the training set. In this way we are able to impose inductive bias beyond what is available in the training data. We investigate two cases:

- MNIST [29]: We use 100 examples of the 0 digits to tune the prior. The training set consists of examples of 1-9 digits, whereas the test set contains 10 000 instances of all digits. We aim to demonstrate the ability of our approach to incorporate prior knowledge about completely unseen data with different characteristics into the model.
- FREY-YALE [12]: We use 1 956 examples of FREY faces to optimize the prior. The training set and test set are comprised of YALE faces. We demonstrate the benefit of using a different dataset but from the same domain (e.g. face images) to specify the prior distribution.

**Visual inspection.**    Fig. 2 shows some qualitative results (additional images are available in the Appendix), while Fig. 3 shows the convergence of the Wasserstein distance during prior optimization in our proposal. From a visual inspection we see that, on MNIST, by encoding knowledge about the "0" digit into the prior, the BAE can reconstruct this digit fairly well although we only use "1" to "9" digits for inference (differently from the BAE with standard prior). Similarly, on FREY-YALE, we see that by encoding knowledge from another dataset in the same domain, the optimized prior can impose a softer constraint compared to using directly this dataset for inference. In addition, if we use directly the union of FREY and YALE faces for training (methods denoted with a ★), VAE yields images that are similar to FREY instead of YALE faces, while generated images from BAE with $\mathcal{N}(0, 1)$ prior are of lower quality. This again highlights the advantage of our approach to specifying an informative prior compared to using that data for training. Another important benefit of our Bayesian treatment of AEs is that we can quantify the *uncertainty* for both reconstructed and generated images. The last

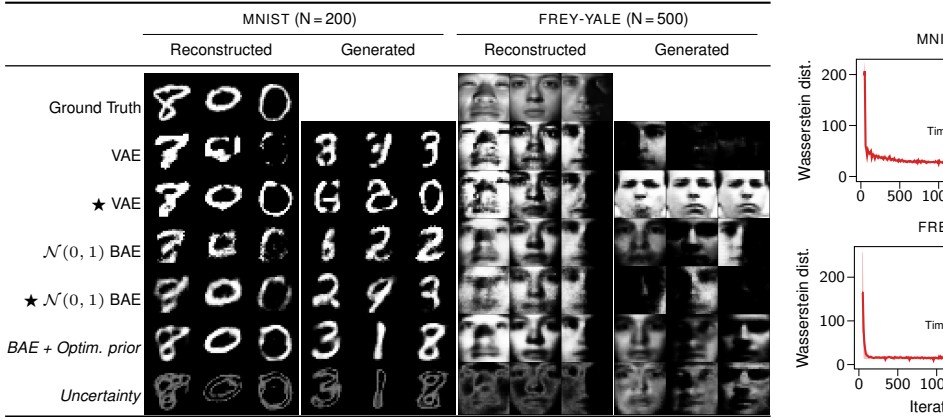

**Figure 2:** Qualitative evaluation for MNIST and YALE. Here, ★ indicates using the union of the training data and the data used to optimize prior to train the model. The last row depicts standard deviation of reconstructed/generated images estimated by BAE using the optimized prior.

**Figure 3:** Convergence of the proposed Wasserstein minimization scheme.

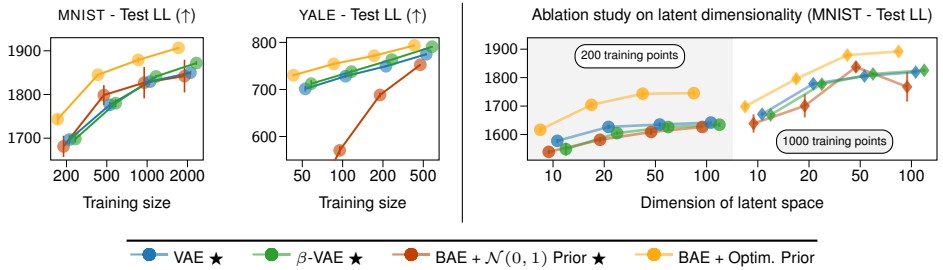

**Figure 4:** Test log-likelihood (LL) of MNIST and YALE. *Left:* test LL as a function of training size; *Right:* test LL as a function of latent dimensionality.

row of Fig. 2 illustrates the uncertainty estimate corresponding to the BAE with optimized prior on MNIST and YALE datasets. Our model exhibits increased uncertainty for semantically and visually challenging pixels such as the left part of the second "0" digit image in the MNIST example. We also observe that the uncertainty is greater for generated images compared to reconstructed images as illustrated in the YALE example. This is reasonable because the reconstruction process is guided by the input data rather than synthesizing new data according to a random latent code.

**Visualization of inductive bias on MNIST.** To have an intuition of the inductive bias induced by the optimized prior, we visualize a low-dimensional projection of parameters sampled from the prior and the posterior [22]. As we see in Fig. 5, the hypothesis space induced by the $\mathcal{N}(0, 1)$ prior is huge, compared to where the true solution should lie. Effectively this is another visualization of the famous Bayesian Occam's razor plot by David MacKay [35], where the model has very high complexity and poor inductive biases. On the other hand, by considering our proposal to do *model selection*, the hypothesis space of the optimized prior is reduced to regions close to the full posterior. Additional visualizations are available in the Appendix.

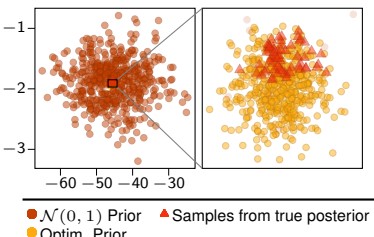

**Figure 5:** Visualization in 2D of samples from priors and posteriors of BAE's parameters. The setup is the same as before with MNIST.

**Quantitative evaluation.** For a quantitative analysis we rely on Fig. 4, where we study the effect on the reconstruction quality of different training sizes (on the *left*) and different latent dimensions (on the *right*). Since we observed that the results of VAE variants are not significantly different, we only show the results for $\beta$-VAE and we leave the extended results to the Appendix. From this experiment we can draw important conclusions. The BAE with optimized priors clearly outperforms

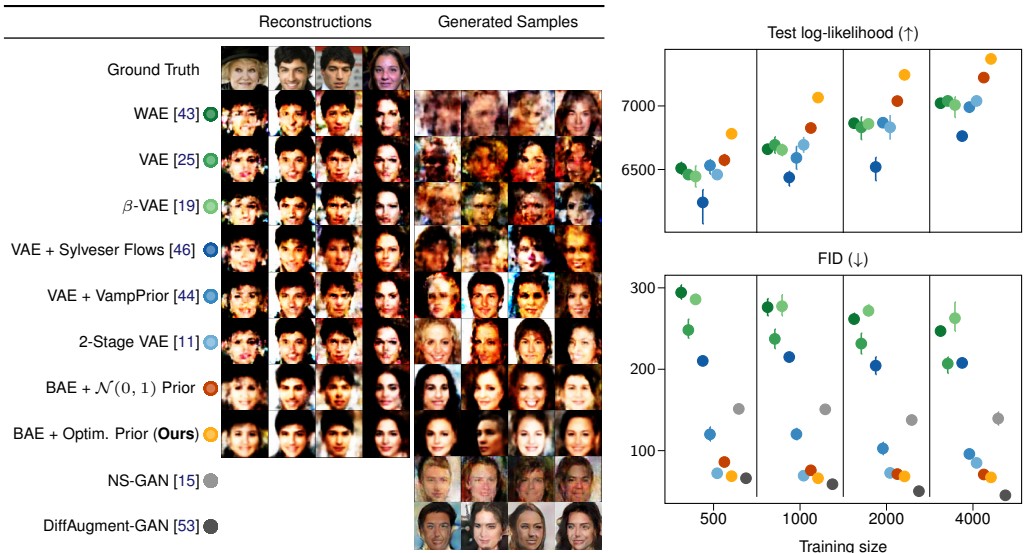

**Figure 6:** Qualitative (*left*) and quantitative evaluation (*right*) on CELEBA. The markers and bars represent the means and one standard deviations, respectively. In the (*left*) figure, the sizes of training data and the data for optimizing prior are 500 and 1000, respectively. The higher the log-likelihood (LL) and the lower FID the better.

the competing methods (and the BAE with standard prior) in the inference task for all training sizes, with slightly diminishing effect for larger sets, as expected. This pattern also holds when looking at different latent dimensions (Fig. 4, *right*), where regardless of the dimensionality of the latent space, BAEs with optimized priors achieve the best performance.

## 4.2 Reconstruction and generation of CELEBA

We now look at a more challenging benchmark, the CELEBA dataset [30]. For our proposal, we use 1 000 examples that are randomly chosen from the original training set to learn the prior distribution. The test set consists of about 20 000 images. The goal of this experiment is to evaluate whether sacrificing part of the training data to specify a good prior is beneficial when compared to using that data for training the model. Fig. 6 shows qualitative results for the competing methods, their corresponding test LLs and FIDs for different training dataset sizes. In terms of test log-likelihoods (LLs) (Fig. 6, *top right*), we observe two clear patterns: (i) that BAE approaches perform considerably better than other methods and (ii) the VAE with Sylvester flows performs consistently poor across dataset sizes. This latter observation indicates that having a more expressive posterior for the encoder is not helpful when considering the small training sizes used in our experiments. More importantly, we see that the BAE using the optimized prior significantly outperforms other methods despite using less data for inference. These results largely agree with the quality of the reconstructions (first column of images in Fig. 6, *left*) in that BAE methods provide more visually appealing reconstructions when compared to other approaches.

We now evaluate the quality of the generated images (second column of images in Fig. 6, *left*) along with their FID scores [18]. Visually, it is clear that images generated from VAEs (standard, $\beta$, Sylvester and Wasserstein Autoencoder (WAE)) are very poor. This failure may originate from the fact that the aggregated posterior distribution of the encoder is not aligned with the prior on the latent space. This problem is more prominent in the case of small training data, where the encoder is not well-trained. The VampPrior tackles this problem by explicitly modeling the aggregated posterior, while 2-stage VAE uses another VAE to estimate the density of the learned latent space. By reducing the effect of the aggregated posterior mismatch, these strategies improve the quality of the generated images remarkably. These results are consistent with their corresponding FID scores (Fig. 6, *bottom right*) where we also see that BAE using the optimized prior consistently outperforms all variants of VAEs and NS-GAN. Finally, we see that DiffAugment-GAN, with the exception of using a training size of 500, yields better FID scores. However, this is not surprising as this model uses much more complex network architectures [24], combined with a powerful differentiable augmentation scheme.

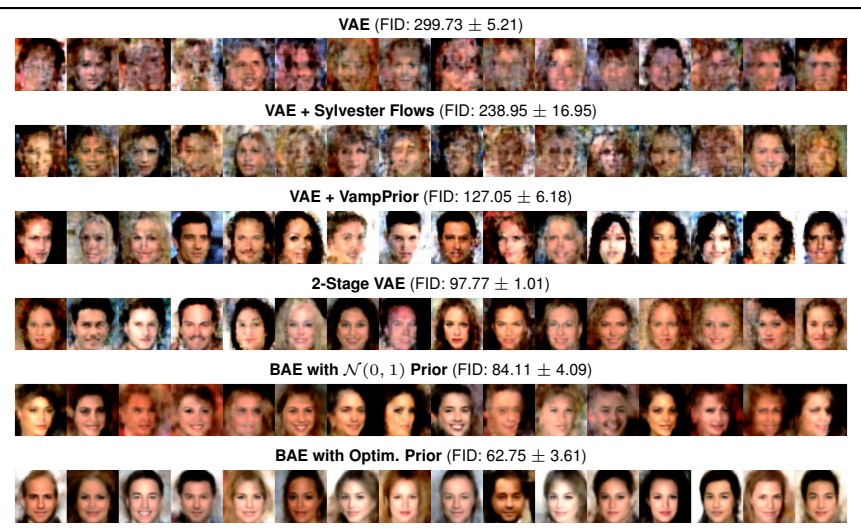

**Figure 7:** Qualitative and quantitative evaluation of generated samples with the *truncated Gaussian likelihood* [8]. Here, we use 500 CELEBA samples for inference.

More importantly, it is clear that with few training samples our method generates more semantically meaningful images then all other approaches, including DiffAugment-GAN.

**The effect of the likelihood.** As previously mentioned, in the Bayesian paradigm, the likelihood represents a modeling choice to be made in addition to the choice of the prior. Our empirical evaluation was predominantly conducted with the continuous Bernoulli likelihood. This likelihood maybe not be an ideal choice for colored images because it biases pixel values to the extremes, which results in saturated images. For the CELEBA dataset, we additionally consider the truncated Gaussian likelihood [8], which is another valid alternative for [0, 1]-valued data (results summarized in Fig. 7). We observe that indeed the colors in the images generated by this likelihood are more realistic and less saturated compared to those generated by the continuous Bernoulli. Still the problem of selecting a good prior is present, as it can be seen for methods like e.g. VAE. For our proposed approach we haven't made strong assumptions on the likelihood (just the ability to sample from it) and as such it is flexible and not tied to a specific choice. These results show that our method is not only quantitatively but also qualitatively better than the competing approaches and confirm that the benefits from our framework are independent of the choice of likelihood. In Appendix, we show more qualitative and numerical results of reconstruction and generation with the truncated Gaussian likelihood.

### 4.3 Prior adjustment versus posterior tempering

We have shown that the proposed framework for adjusting the prior is compatible with standard Bayesian practices, as it emulates type-II maximum likelihood. In other words, the distribution fitting that we induce by means of Wasserstein distance minimization relates to the marginal output of BAEs, very much in the same spirit of marginal likelihood maximization. The distribution is fit considering *all* possible functions, when marginalized through the likelihood, creating an implicit regularization effect. Our scheme does not give more weight to particular training instances, but it simply restricts the hypothesis space. This is unlike *posterior tempering* [51, 22, 48, 1, 50], which is commonly defined as $p_\tau(\mathbf{w} \mid \mathbf{y}) \propto p(\mathbf{y} \mid \mathbf{w})^{1/\tau} p(\mathbf{w})$, where $\tau > 0$ is a *temperature* value. With $\tau < 1$, tempering is known to improve performance in the case of small training data and using miss-specified priors, but it corresponds to artificially sharpening the posterior by over-counting the data $\tau$ times.

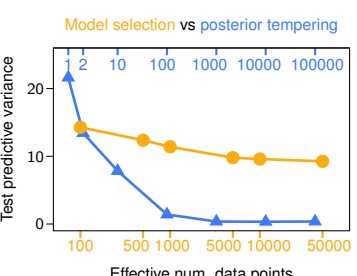

**Figure 8:** Average test predictive variance as a function of the number of data points used to optimize the prior, and the temperature (i.e. how many times the data points are over-counted).

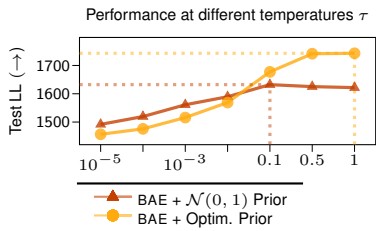

**Figure 9:** Test performance for temperature scaling with different priors. The dotted lines indicate the best performance.

To demonstrate the differences with our proposal, we setup a comparison on MNIST. In the empirical comparison of Fig. 8, we consider different temperatures and different sets of data points used to optimize the prior. As expected, the tempered posterior quickly collapses on the mode, while the posterior after our treatment retains a sufficiently constant variance, regardless of the number of data points used. It is also interesting to notice that with the $\mathcal{N}(0, 1)$ prior, the best temperature is $\tau = 0.1$, while for our approach that optimizes the prior is $\tau = 1$, further confirming that the model now is well specified (Fig. 9).

## 5 Conclusions

In this work, we have reconsidered the Bayesian treatment of autoencoders (AE) in light of recent advances in Bayesian neural networks. We have addressed the main challenge of BAEs, so that they can be rendered as viable alternative to generative models such as VAEs. More specifically, we have found that the main limitation of BAEs lies in the difficulty of specifying meaningful priors in the context of highly-structured data, which is ubiquitous in modern machine learning applications. Consequently, we have proposed to specify priors over the autoencoder weights by means of a novel optimization of prior hyper-parameters. Inspired by connections with marginal likelihood optimization, we derived a practical and efficient optimization framework, based on the minimization of the distributional sliced-Wasserstein distance between the distribution induced by the BAE and the data generating distribution. The resulting hyper-parameter optimization strategy leads to a novel way to perform model selection for BAEs, and we showed its advantages in an extensive experimental campaign.

**Limitations and ethical concerns.** Even if theoretically justified and empirically verified with extensive experimentation, our proposal for model selection still remains a *proxy* to the true marginal likelihood maximization. The DSWD formulation has nice properties of asymptotic convergence and computational tractability, but it may represent only one of the possible solutions. At the same time, we stress that the current literature does not cover this problem of BAEs at all, and we believe our approach is a considerable step towards the development of practical Bayesian methods for representation learning in modern applications characterized by large-scale structured data (including tabular and graph data, which are currently not covered). At the same time, the accessibility to these models to a wider audience and different kind of data might help to widespread harmful applications, which is a concern shared among all generative modeling approaches. An ethical analysis of the consequences of Bayesian priors in unsupervised learning scenarios is also worth an in-depth investigation, which goes beyond the scope of this work.

## Acknowledgments and Disclosure of Funding

MF gratefully acknowledges support from the AXA Research Fund and the Agence Nationale de la Recherche (grant ANR-18-CE46-0002 and ANR-19-P3IA-0002).

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
