# Model Selection for Bayesian Autoencoders: Supplementary Material

**Ba-Hien Tran**
EURECOM
(France)

**Simone Rossi**
EURECOM
(France)

**Dimitrios Milios**
EURECOM
(France)

**Pietro Michiardi**
EURECOM
(France)

**Edwin V. Bonilla**
CSIRO's Data61
The Australian National University
The University of Sydney
(Australia)

**Maurizio Filippone**
EURECOM
(France)

## A    Derivation of Distributional Sliced-Wasserstein Distance

In this section, we review some key results on the Wasserstein distance. Given two probability measures $\pi$, $\rho$, both defined on $\mathbb{R}^D$ for simplicity, the $p$-Wasserstein distance between $\pi$ and $\rho$ is given by

$$W_p^p(\pi, \rho) = \inf_{\gamma \in \Gamma(\pi, \rho)} \int \|\mathbf{x} - \mathbf{y}\|^p \gamma(\mathbf{x}, \mathbf{y}) \mathrm{d}\mathbf{x}\mathrm{d}\mathbf{y}\,, \tag{1}$$

where $\Gamma(\pi, \rho)$ is the set of all possible distributions $\gamma(\mathbf{x}, \mathbf{y})$ such that the marginals are $\pi(\mathbf{x})$ and $\rho(\mathbf{y})$ [32]. While usually analytically unavailable, for $D = 1$ the distance has the following closed form solution,

$$W_p^p(\pi, \rho) = \int_0^1 |F_\pi^{-1}(z) - F_\rho^{-1}(z)|^p \mathrm{d}z\,, \tag{2}$$

where $F_\pi$ and $F_\rho$ are the cumulative density functions (CDFs) of $\pi$ and $\rho$, respectively.

### A.1    (Distributional) Sliced-Wasserstein Distance

The main idea underlying the distributional sliced-Wasserstein distance (DSWD) is to project the challenging estimation of distances for high-dimensional distributions into simpler estimation of multiple distances in one dimension, which all have closed-form solution (Eq. 2). The projection is done using the Radon transform $\mathcal{R}$, an operator that maps a density function $\varphi$ defined in $\mathbb{R}^D$ to the set of its integrals over hyperplanes in $\mathbb{R}^D$,

$$\mathcal{R}\varphi(t, \boldsymbol{\theta}) := \int \varphi(\mathbf{z})\delta(t - \mathbf{z}^\top \boldsymbol{\theta})\mathrm{d}\mathbf{z}, \quad \forall t \in \mathbb{R}, \ \ \forall \boldsymbol{\theta} \in \mathbb{S}^{D-1}\,, \tag{3}$$

where $\mathbb{S}^{D-1}$ is the unit sphere in $\mathbb{R}^D$ and $\delta(\cdot)$ is the Dirac delta [11]. Using the Radon transform, for a given $\boldsymbol{\theta}$ we can project the two densities $\pi$ and $\rho$ into one dimension,

$$W_p^p(\pi, \rho) = \int_{\mathbb{S}^{D-1}} W_p^p\left(\mathcal{R}\pi(t, \boldsymbol{\theta}), \mathcal{R}\rho(t, \boldsymbol{\theta})\right) \mathrm{d}\boldsymbol{\theta} \approx \frac{1}{K} \sum_{i=1}^K W_p^p\left(\mathcal{R}\pi(t, \boldsymbol{\theta}_i), \mathcal{R}\rho(t, \boldsymbol{\theta}_i)\right), \tag{4}$$

where the approximation comes from using Monte-Carlo integration by sampling $\boldsymbol{\theta}_i$ uniformly in $\mathbb{S}^{D-1}$ [2]. While having significant computational advantages, this approach might require to

35th Conference on Neural Information Processing Systems (NeurIPS 2021).

draw many unimportant projections that are computationally exhausting and that provide a minimal improvement on the overall distance approximation.

The *distributional sliced-Wasserstein distance* (DSW) [25] solves this issue by finding the optimal probability measure of slices $\sigma(\boldsymbol{\theta})$ on the unit sphere $\mathbb{S}^{D-1}$ and it's defined as follows,

$$DSW_p(\pi, \rho; C) := \sup_{\sigma \in \mathbb{M}_C} \left( \mathbb{E}_{\sigma(\boldsymbol{\theta})} W_p^p\big(\mathcal{R}\pi(t, \boldsymbol{\theta}), \mathcal{R}\rho(t, \boldsymbol{\theta})\big) \right)^{1/p}, \tag{5}$$

where, for $C > 0$, $\mathbb{M}_C$ is the set of probability measures $\sigma$ such that $\mathbb{E}_{\boldsymbol{\theta}, \boldsymbol{\theta}' \sim \sigma}\big[\boldsymbol{\theta}^\top \boldsymbol{\theta}'\big] \leq C$ (a constraint that aims to avoid directions to lie in only one small area). Critically, the definition of DSWD in Eq. 5 does not suffer from the curse of dimensionality, indeed [25] showed that the statistical error of this estimation scales down with $C_D \cdot n^{-\frac{1}{2}}$, where $C_D$ is a constant depending on dimension $D$. Furthermore, while generally we have that $DSW_p(\pi, \rho) \leq W_p(\pi, \rho)$, it can be proved that under mild assumptions on $C$, the two distances are topological equivalent, i.e. converging in distribution on $DSW_p$ implies the convergence on $W_p$ [see Theorem 2 in 25].

The direct computation of $DSW_p$ in Eq. 5 is still challenging but it admits an equivalent dual form,

$$\sup_{h \in \mathcal{H}} \left\{ \left( \mathbb{E}_{\bar{\sigma}(\boldsymbol{\theta})}\big[ W_p^p\big(\mathcal{R}\pi(t, h(\boldsymbol{\theta})), \mathcal{R}\rho(t, h(\boldsymbol{\theta}))\big)\big] \right)^{1/p} - \lambda_C \mathbb{E}_{\boldsymbol{\theta}, \boldsymbol{\theta}' \sim \bar{\sigma}}\big[\big|h(\boldsymbol{\theta})^\top h(\boldsymbol{\theta}')\big|\big] \right\} + \lambda_C C, \tag{6}$$

where $\bar{\sigma}$ is a uniform distribution in $\mathbb{S}^{D-1}$, $\mathcal{H}$ is a class of all Borel measurable functions $\mathbb{S}^{D-1} \rightarrow \mathbb{S}^{D-1}$ and $\lambda_C$ is a regularization hyper-parameter. The formulation in Eq. 6 is obtained by employing the Lagrangian duality theorem and by reparameterizing $\sigma(\boldsymbol{\theta})$ as push-forward transformation of a uniform measure in $\mathbb{S}^{D-1}$ via $h$. Now, by parameterizing $h$ using a deep neural network[1] with parameters $\phi$, defined as $h_\phi$, Eq. 6 becomes an optimization problem with respect to the network parameters. The final step is to approximate the analytically intractable expectations with Monte Carlo integration,

$$DSW_p(\pi, \rho) \approx$$

$$\max_{\phi} \left\{ \left[ \frac{1}{K} \sum_{i=1}^{K} \big[ W_p^p\big(\mathcal{R}\pi(t, h_\phi(\boldsymbol{\theta}_i)), \mathcal{R}\rho(t, h_\phi(\boldsymbol{\theta}_i))\big)\big] \right]^{1/p} - \frac{\lambda_C}{K^2} \sum_{i,j=1}^{K} |h_\phi(\boldsymbol{\theta}_i)^\top h_\phi(\boldsymbol{\theta}_j)| + \lambda_C C \right\}, \tag{7}$$

where $\boldsymbol{\theta}_i$ are uniform samples from the unit sphere $\mathbb{S}^{D-1}$ and $\forall t \in \mathbb{R}$. Finally, we can use stochastic gradient methods to update $\phi$ and then use the resulting optima for the estimation of the original distance.

## B    Numerical Implementation of Sliced-Wasserstein Distance

### B.1    Wasserstein distance between two empirical 1D distributions

The Wasserstein distance between two one-dimensional distributions $\pi$ and $\rho$ is defined as in Eq. 2. The integral in this equation can be numerically estimated by using the midpoint Riemann sum:

$$\int_0^1 |F_\pi^{-1}(z) - F_\rho^{-1}(z)|^p dz \approx \frac{1}{M} \sum_{m=1}^{M} |F_\pi^{-1}(z_m) - F_\rho^{-1}(z_m)|^p, \tag{8}$$

where $z_m = \frac{2m-1}{M}$, $M$ is the number of points used to approximate the integral. If we only have samples from the distributions, $x_m \sim \pi$ and $y_m \sim \rho$, we can obtain the empirical densities as follows

$$\pi(x) \approx \pi_M(x) = \frac{1}{M} \sum_{m=1}^{M} \delta(x - x_m), \tag{9}$$

$$\rho(y) \approx \rho_M(y) = \frac{1}{M} \sum_{m=1}^{M} \delta(y - y_m), \tag{10}$$

---

[1] We use a single multi layer perceptron (MLP) layer with normalized output as the $h$ function.

where $\delta$ is the Dirac delta function. The corresponding empirical cumulative density functions are

$$F_\pi(z) \approx F_{\pi,M}(z) = \frac{1}{M} \sum_{m=1}^{M} u(z - x_m), \tag{11}$$

$$F_\rho(z) \approx F_{\rho,M}(z) = \frac{1}{M} \sum_{m=1}^{M} u(z - y_m), \tag{12}$$

where $M$ is the number of samples, $u(\cdot)$ is the step function.

Calculating the Wasserstein distance with the empirical distribution function is computationally attractive. To do that, we first sort $x_m s$ in an ascending order, such that $x_{i[m]} \leq x_{i[m+1]}$, where $i[m]$ is the index of the sorted $x_m s$. It is straightforward to show that $F_{\pi,M}^{-1}(z_m) = x_{i[m]}$. Thus, the Wasserstein distance can be approximated as follows

$$W_p^p(\pi, \rho) \approx \frac{1}{M} \sum_{m=1}^{M} |x_{i[m]} - y_{j[m]}|^p. \tag{13}$$

### B.2 Slicing empirical distribution

According to the equation Eq. 3, the marginal densities (i.e. slices) of the distribution $\pi$ can be obtained as follows

$$\mathcal{R}\pi(t, \boldsymbol{\theta}) = \int \pi(\mathbf{x})\delta(t - \mathbf{x}^\top \boldsymbol{\theta})d\mathbf{x}, \quad \forall t \in \mathbb{R}. \tag{14}$$

Because, in practice, only samples from the distributions are available we aim to calculate a Radon slice of the empirical distribution of $M$ samples $\pi_M = \frac{1}{M} \sum_{m=1}^{M} \delta(\mathbf{x} - \mathbf{x}_m)$:

$$\mathcal{R}\pi(t, \boldsymbol{\theta}) \approx \frac{1}{M} \sum_{m=1}^{M} \int \delta(\mathbf{x} - \mathbf{x}_m)\delta(t - \mathbf{x}^\top \boldsymbol{\theta})d\mathbf{x} \tag{15}$$

$$= \frac{1}{M} \sum_{m=1}^{M} \delta(t - \mathbf{x}_m^\top \boldsymbol{\theta}). \tag{16}$$

By using the approximation in Eq. 16 and the empirical implementation of 1D Wasserstein distance (Eq. 13), we are able to compute a proxy to the original distance in Eq. 5.

## C  Pseudocode of Prior Optimization Procedure

Algorithm 1 desribes the procedure of prior optimization for Bayesian autoencoders (BAEs).

---

**Algorithm 1:** Prior Optimization

**Input:** Empirical distribution $\tilde{\pi}(\mathbf{y})$; prior over parameters $p_{\boldsymbol{\psi}}(\mathbf{w})$; number of prior samples $N_S$; mini-batch size $N_B$; number of random projections $K$; regularization coefficient $\lambda_C$.
**Output:** The optimized prior's parameters $\boldsymbol{\psi}$

1 **while** *$\boldsymbol{\psi}$ has not converged* **do**
2    Sample $\mathbf{x} \overset{\text{def}}{=} \mathbf{y} = \{\mathbf{y}_i\}_{i=1}^{N_B}$ from $\tilde{\pi}(\mathbf{y})$ // `Sample input data`
3    Sample $\mathcal{W} = \{\mathbf{w}_i\}_{i=1}^{N_S}$ from $p_{\boldsymbol{\psi}}(\mathbf{w})$ // `Sample parameters from the prior`
4    **foreach** $\mathbf{w}_i \in \mathcal{W}$ **do**
        /* Following steps are performed in a batch manner                */
5       $\mathbf{f}_i = (f_{\text{dec}} \circ f_{\text{enc}})(\mathbf{x})$ // `Compute the functional outputs from Autoencoder`
6       Sample $\tilde{\mathbf{y}}_i$ from $p(\mathbf{y} | \mathbf{f}_i)$ // `Sample from the likelihood`
7    Gather samples $\tilde{\mathbf{y}} = \cup \{\tilde{\mathbf{y}}_i\}_{i=1}^{N_s}$
8    $\mathcal{L} = DSW_2(\mathbf{y}, \tilde{\mathbf{y}}; K, \lambda_C)$ // `Compute the` $DSW_2$ `distance using` Eq. 7
9    $\boldsymbol{\psi} \leftarrow \text{Optimizer}(\boldsymbol{\psi}, \nabla_{\boldsymbol{\psi}}\mathcal{L})$ // `Update prior's parameters`
10 **Return:** $\boldsymbol{\psi}$

---

# D   Details on Stochastic gradient Hamiltonian Monte Carlo

Hamiltonian Monte Carlo (HMC) [24] is a highly-efficient Markov Chain Monte Carlo (MCMC) method used to generate samples from the posterior $\mathbf{w} \sim p(\mathbf{w} \mid \mathbf{y})$. HMC considers the joint log-likelihood as a pontential energy function $U(\mathbf{w}) = -\log p(\mathbf{y} \mid \mathbf{w}) - \log p(\mathbf{w})$, and introduces a set of auxilary momentum variable $\mathbf{r}$. Samples are generated from the joint distribution $p(\mathbf{w}, \mathbf{r})$ based on the Hamiltonian dynamics:

$$\begin{cases} d\mathbf{w} & = \mathbf{M}^{-1}\mathbf{r}dt, \\ d\mathbf{r} & = -\nabla U(\mathbf{w})dt, \end{cases} \tag{17}$$

where, $\mathbf{M}$ is an arbitrary mass matrix that plays the role of a preconditioner. In practice, this continuous system is approximated by means of $\varepsilon$-discretized numerical integration, and followed by Metropolis steps to accommodate numerical errors stemming from the integration.

However, HMC is not practical for large datasets due to the cost of computing the gradient $\nabla U(\mathbf{w}) = \nabla \log(\mathbf{y} \mid \mathbf{w})$ on the entire dataset. To mitigate this issue, [4] proposed stochastic gradient Hamiltonian Monte Carlo (SGHMC), which uses a noisy, unbiased estimate of the gradient $\nabla \tilde{U}(\mathbf{w})$ which is computed from a mini-batch of the data. The discretized Hamiltonian dynamics are then updated as follows

$$\begin{cases} \Delta\mathbf{w} & = \varepsilon\mathbf{M}^{-1}\mathbf{r}, \\ \Delta\mathbf{r} & = -\varepsilon\nabla\tilde{U}(\mathbf{w}) - \varepsilon\mathbf{C}\mathbf{M}^{-1}\mathbf{r} + \mathcal{N}(0, 2\varepsilon(\mathbf{C} - \tilde{\mathbf{B}})), \end{cases} \tag{18}$$

where $\varepsilon$ is an step size, $\mathbf{C}$ is an user-defined friction matrix, $\tilde{\mathbf{B}}$ is the estimate for the noise of the gradient evaluation. To choose these hyper-parameters, we use a scale-adapted version of SGHMC [28], where the hyper-parameters are adjusted automatically during a burn-in phase. After this period, all hyperparamteters stay fixed.

**Estimating M.**   We set the mass matrix $\mathbf{M}^{-1} = \mathrm{diag}\left(\hat{V}_\mathbf{w}^{-1/2}\right)$, where $\hat{V}_\mathbf{w}$ is an estimate of the uncentered variance of the gradient, $\hat{V}_\mathbf{w} \approx \mathbb{E}[(\nabla\tilde{U}(\mathbf{w}))^2]$, which can be estimated by using exponential moving average as follows

$$\Delta\hat{V}_\mathbf{w} = -\tau^{-1}\hat{V}_\mathbf{w} + \tau^{-1}\nabla(\tilde{U}(\mathbf{w}))^2, \tag{19}$$

where $\tau$ is a parameter vector that specifies the moving average windows. This parameter can be automatically chosen by using an adaptive estimate [28] as follows

$$\Delta\tau = -g_\mathbf{w}^2\hat{V}_\mathbf{w}^{-1}\tau + 1, \quad \text{and,} \quad \Delta g_\mathbf{w} = -\tau^{-1}g_\mathbf{w} + \tau^{-1}\nabla\tilde{U}(\mathbf{w}), \tag{20}$$

where $g_\mathbf{w}$ is a smoothed estimate of the gradient $\nabla U(\mathbf{w})$.

**Estimating $\tilde{\mathbf{B}}$.**   The estimate for the noise of the gradient evaluation, $\tilde{\mathbf{B}}$ should be ideally the estimate of empirical Fisher information matrix of $U(\mathbf{w})$, which is prohibitively expensive to compute. Therefore, we use a diagonal approximation, $\tilde{\mathbf{B}} = \frac{1}{2}\varepsilon\hat{V}_\mathbf{w}$, which is already available from the step of estimating $\mathbf{M}$.

**Choosing C.**   In practice, one can simply set the friction matrix as $\mathbf{C} = C\mathbf{I}$, i.e. the same independent noise for each elements of $\mathbf{w}$.

**The discretized Hamiltonian dynamics.**   By substituting $\mathbf{v} := \varepsilon\hat{V}_\mathbf{w}^{-1/2}\mathbf{r}$, the dynamics Eq. 18 become

$$\begin{cases} \Delta\mathbf{w} & = \mathbf{v}, \\ \Delta\mathbf{v} & = -\varepsilon^2\hat{V}_\mathbf{w}^{-1/2}\nabla\tilde{U}(\mathbf{w}) - \varepsilon C\hat{V}_\mathbf{w}^{-1/2}\mathbf{v} + \mathcal{N}(0, 2\varepsilon^3 C\hat{V}_\mathbf{w}^{-1} - \varepsilon^4\mathbf{I}). \end{cases} \tag{21}$$

Following [28], we choose $C$ such that $\varepsilon C\hat{V}_\mathbf{w}^{-1/2} = \alpha\mathbf{I}$. This is equivalent to using a constant momentum coefficient of $\alpha$. The final discretized dynamics are then

$$\begin{cases} \Delta\mathbf{w} & = \mathbf{v}, \\ \Delta\mathbf{v} & = -\varepsilon^2\hat{V}_\mathbf{w}^{-1/2}\nabla\tilde{U}(\mathbf{w}) - \alpha\mathbf{v} + \mathcal{N}(0, 2\varepsilon^2\alpha\hat{V}_\mathbf{w}^{-1/2} - \varepsilon^4\mathbf{I}). \end{cases} \tag{22}$$

# E PCA of the SGD Trajectory

Inspired by [14], we use the subspace spanned by the SGD trajectory to visualize neural network's parameters in a low-dimensional space. This subspace is cheap to construct and can capture many of the sharp directions of the loss surface [14, 20, 23]. More specifically, we perform SGD starting from a MAP solution with a constant learning rate. Here, the loss function is the negative log joint likelihood of the BAE:

$$\mathcal{L}(\mathbf{w}) = -\frac{N}{M} \sum_{i=1}^{M} \log p(\mathbf{y}_i \mid \mathbf{w}) - \log p(\mathbf{w}), \tag{23}$$

where $M$ is the mini-batch size and $N$ is the size of training data. We store the deviations $\mathbf{a}_i = \overline{\mathbf{w}} - \mathbf{w}_i$ for the last $M$ epochs, where $\overline{\mathbf{w}}$ is the running average of the first moment, $M$ is determined by the amount of memory we can use. Then we perform PCA based on randomized SVD [10] on the matrix $\mathbf{A}$ comprised of vectors $\mathbf{a}_1, ..., \mathbf{a}_M$ to construct the subspace. The procedure is summarized in Algorithm 2.

---

**Algorithm 2:** Subspace construction with PCA

---

**Input:** Pretrained paremeters $\mathbf{w}_{\text{MAP}}$; learning rate $\eta$; number of steps $\tau$; momentum update frequency $c$; maximum number of columns $M$ in deviation matrix $\mathbf{A}$.

**Output:** Shift vector $\overline{\mathbf{w}}$; projection matrix $\mathbf{P}$ for subspace.

1   $\overline{\mathbf{w}} \leftarrow \mathbf{w}_{\text{MAP}}$ // Initialize mean
2   **for** $i \leftarrow 1, 2, ..., T$ **do**
3     $\mathbf{w}_i \leftarrow \mathbf{w}_{i-1} - \eta \nabla_{\mathbf{w}} \mathcal{L}(\mathbf{w}_{i-1})$ // Perform SGD update
4     **if** $\text{MOD}(i, c) = 0$ **then**
5       $n \leftarrow i/c$ // Number of models
6       $\overline{\mathbf{w}} \leftarrow \frac{n\overline{\mathbf{w}} + \mathbf{w}_i}{n+1}$ // Update mean
7       **if** $\text{NUM\_COLS}(\mathbf{A}) = M$ **then**
8         $\text{REMOVE\_COL}(\mathbf{A}[:, 1])$
9       $\text{APPEND\_COL}(\mathbf{A}, \mathbf{w}_i - \overline{\mathbf{w}})$ // Store deviation

10   $\mathbf{U}, \mathbf{S}, \mathbf{V}^\top \leftarrow SVD(\mathbf{A})$ // Perform truncated SVD
11   **Return:** $\overline{\mathbf{w}}, \mathbf{P} = \mathbf{S}\mathbf{V}^\top$

---

# F  Additional Details on Experimental Settings

## F.1  Experimental environment

In our experiments, we use 4 workstations, which have the following specifications:

- **GPU**: NVIDIA Tesla P100 PCIe 16 GB.
- **CPU**: Intel(R) Xeon(R) (4 cores) @ 2.30GHz.
- **Memory**: 25.5 GiB (DDR3).

## F.2  Preprocessing data

- MNIST [19]: The dataset is publicly available at `http://yann.lecun.com/exdb/mnist`. We keep the original resolution of $1 \times 28 \times 28$ of the MNIST dataset.
- FREY-YALE [6]:  The FREY and YALE datasets are publicly availaibe at `http://cs.nyu.edu/~roweis/data.html` and `http://vision.ucsd.edu/extyaleb/CroppedYaleBZip`, respectivey.  All the images of FREY and YALE datasets are resized to the $1 \times 28 \times 28$ resolution.
- CELEBA [21]: The dataset is publicly available at `http://mmlab.ie.cuhk.edu.hk/projects/CelebA.html`. According to [7], we pre-process CELEBA images by first taking a $148 \times 148$ center crop and then resizing to the $3 \times 64 \times 64$ resolution.

## F.3  Network architectures

In our experiments, we use convolutional networks for modeling both encoders and decoders. For a fair comparison, we employ the same network architecture for all models. The network's parameters are initialized by using the default scheme in PyTorch [27].

Table 1 shows details on the network architectures used in our experimental campaign.

|  | MNIST | FREY-YALE | CELEBA |
|---|---|---|---|
| ENCODER: | $x \in \mathbb{R}^{1\times 28\times 28}$ | $x \in \mathbb{R}^{1\times 28\times 28}$ | $x \in \mathbb{R}^{3\times 64\times 64}$ |
|  | $\rightarrow$ CONV$_{32}$ $\rightarrow$ LEAKY RELU | $\rightarrow$ CONV$_{64}$ $\rightarrow$ LEAKY RELU | $\rightarrow$ CONV$_{64}$ $\rightarrow$ LEAKY RELU |
|  | $\rightarrow$ CONV$_{64}$ $\rightarrow$ LEAKY RELU | $\rightarrow$ CONV$_{128}$ $\rightarrow$ LEAKY RELU | $\rightarrow$ CONV$_{128}$ $\rightarrow$ LEAKY RELU |
|  | $\rightarrow$ CONV$_{64}$ $\rightarrow$ LEAKY RELU | $\rightarrow$ CONV$_{128}$ $\rightarrow$ LEAKY RELU | $\rightarrow$ CONV$_{256}$ $\rightarrow$ LEAKY RELU |
|  | $\rightarrow$ CONV$_{128}$ $\rightarrow$ LEAKY RELU | $\rightarrow$ CONV$_{256}$ $\rightarrow$ LEAKY RELU | $\rightarrow$ CONV$_{512}$ $\rightarrow$ LEAKY RELU |
|  | $\rightarrow$ FLATTEN $\rightarrow$ FC$_{50\times M}$ | $\rightarrow$ FLATTEN $\rightarrow$ FC$_{50\times M}$ | $\rightarrow$ FLATTEN $\rightarrow$ FC$_{50\times M}$ |
| DECODER: | $z \in \mathbb{R}^{50} \rightarrow$ FC$_{7\times 7\times 128}$ | $z \in \mathbb{R}^{50} \rightarrow$ FC$_{7\times 7\times 256}$ | $z \in \mathbb{R}^{50} \rightarrow$ FC$_{8\times 8\times 512}$ |
|  | $\rightarrow$ LEAKY RELU | $\rightarrow$ LEAKY RELU | $\rightarrow$ LEAKY RELU |
|  | $\rightarrow$ CONVT$_{128}$ $\rightarrow$ LEAKY RELU | $\rightarrow$ CONVT$_{256}$ $\rightarrow$ LEAKY RELU | $\rightarrow$ CONVT$_{512}$ $\rightarrow$ LEAKY RELU |
|  | $\rightarrow$ CONVT$_{64}$ $\rightarrow$ LEAKY RELU | $\rightarrow$ CONVT$_{128}$ $\rightarrow$ LEAKY RELU | $\rightarrow$ CONVT$_{256}$ $\rightarrow$ LEAKY RELU |
|  | $\rightarrow$ CONVT$_{64}$ $\rightarrow$ LEAKY RELU | $\rightarrow$ CONVT$_{128}$ $\rightarrow$ LEAKY RELU | $\rightarrow$ CONVT$_{128}$ $\rightarrow$ LEAKY RELU |
|  | $\rightarrow$ CONVT$_{1}$ $\rightarrow$ SIGMOID | $\rightarrow$ CONVT$_{1}$ $\rightarrow$ SIGMOID | $\rightarrow$ CONVT$_{1}$ $\rightarrow$ SIGMOID |

**Table 1:** Convolutional Encoder-Decoder architectures. CONV$_n$ denotes a convolutional layer with $n$ filters, whereas FC$_n$ represents a fully-connected layer with $n$ units. All convolutions CONV$_n$ and transposed convolutions CONVT$_n$ have a filter size of $4\times 4$ for MNIST and FREY-YALE and $5\times 5$ for CELEBA. $M = 1$ for all models except for the Variational Autoencoders (VAEs) which have $M = 2$ as the encoder has to yield both mean and variance for each input.

## F.4  Prior optimiziation

As done in [25], we use a single-layer multilayer perceptron (MLP), $h_\phi$, to represent the Borel measurable function in the dual form of DSWD (Eq. 7). At each iteration of Algorithm 1, to find a local maxima, we optimize $h_\phi$ for 30 epochs by using an Adam optimizer [17] with a learning rate of 0.0005. We use another Adam optimizer with a learning rate of 0.001 to update the prior's parameters. We use a mini-batch size of $N_B = 64$ and then generate $N_s = 32$ prior samples given each data point. By default, we use $K = 1000$ random projections with a regularization coefficient $\lambda_C = 100$ to estimate the 2-Wasserstein distance. The convergences of prior optimization on MNIST, FREY and CELEBA datasets are illustrated in Fig. 8.

### F.5 SGHMC hyper-parameters

In Table 2 we report the hyper-parameters used in the experiments on MNIST, YALE and CELEBA datasets. As seen, we always use a fixed step size of $0.003$, a momentum coefficient of $0.05$, and a mini-batch size of $64$. The number of collected samples after thinning is $32$. The number of burn-in iterations and the thinning interval are increased according to the size of the training set.

| | MNIST | | | | YALE | | | | CELEBA | | | |
|---|---|---|---|---|---|---|---|---|---|---|---|---|
| TRAINING SIZE | 200 | 500 | 1000 | 2000 | 50 | 100 | 200 | 500 | 500 | 1000 | 2000 | 4000 |
| MINI-BATCH SIZE | 64 | 64 | 64 | 64 | 64 | 64 | 64 | 64 | 64 | 64 | 64 | 64 |
| STEP SIZE $(10^{-3})$ | 3 | 3 | 3 | 3 | 3 | 3 | 3 | 3 | 3 | 3 | 3 | 3 |
| MOMENTUM $(10^{-2})$ | 5 | 5 | 5 | 5 | 5 | 5 | 5 | 5 | 5 | 5 | 5 | 5 |
| NUM. BURN-IN STEPS $(10^3)$ | 6 | 6 | 6 | 6 | 6 | 6 | 6 | 6 | 6 | 20 | 20 | 20 |
| NUM. SAMPLES | 32 | 32 | 32 | 32 | 32 | 32 | 32 | 32 | 32 | 32 | 32 | 32 |
| THINNING INTERVAL $(10^3)$ | 1 | 1 | 1 | 2 | 1 | 1 | 1 | 1 | 1 | 2 | 3 | 5 |

**Table 2:** SGHMC hyper-parameters used in the experiments on MNIST, YALE and CELEBA datasets.

### F.6 Competing approaches

- **VAE** [18]: The vanilla VAE model employed with a Gaussian encoder and a standard Gaussian prior on the latent space.

- $\beta$-**VAE** [13]: The Kullback-Leibler divergence (KL) term in the VAE's objective is weighted by $\beta = 0.1$ to reduce the effect of the prior. This helps to avoid the over-regularization problem of VAEs and improve reconstruction quality.

- **VAE + Sylvester Flows** [31]: One of the state-of-the-art normalizing flows for the encoder of VAEs, which has richer expressiveness than VAE's post-Gaussian encoder. As employed in [31], we use Orthogonal Sylvester flows with $4$ transformations and $32$ orthogonal vectors.

- **VAE + VampPrior** [30]: A flexible prior for VAEs, which is a mixture of variational posteriors conditioned on learnable pseudo-observations. This allows the variational posterior to learn more a potential latent representation. Due to using small training data, we use $100$ trainable pseudo-observations in our experiments. We found that increasing more pseudo-observations may hurt the predictive performance because of overfitting.

- **2-Stage VAE** [5]: A simple and practical method to improve the quality of generated images from VAEs by performing a form of ex-post density estimation via a second VAE. As employed in [5], for the second-stage VAE, we use a MLP having three 1024-dimensional hidden layers with ReLU activation function.

- **WAE** [29] Wasserstein Autoencoder: This model is an alternative of VAEs. By reformulating the objective function as an optimal transport (OT) problem, Wasserstein Autoencoder (WAE) regularizes the averaged encoding distribution instead of each data point. This encourages the encoded training distribution to match the prior while still allowing to learn significant information from the data. As suggested in [29], we use WAE-MMD with the inverse multiquadratics kernel and a regularization coefficient $\lambda = 10$ due to its stability compared to WAE-GAN. We impose the standard Gaussian prior on the latent space.

- **NS-GAN** [9]: a standard Generative Adversarial Network (GAN) with the non-saturating loss, which has been shown to be robust to the choice of hyper-parameters on CELEBA [22]. For a fair comparison, we reuse the encoder and decoder architectures for the discriminator and generator, respectively.

- **DiffAugment-GAN** [37]: a more complex architecture [STYLEGAN2, see 16] combined with a powerful differentiable augmentation scheme, specifically developed for low data regimes. We refer to the original work of [37] and the implementation in https://github.com/mit-han-lab/data-efficient-gans for additional details on the network architecture. We use the same latent size of 50, a maximum of 64 feature maps, and all available augmentations (color, cutout and translation). The remaining parameters are left at default value.

All autoencoder models are trained for 200 epochs with an Adam optimizer [17] using the default hyper-parameters in PyTorch, i.e. learning rate $= 0.001$, $\beta_1 = 0.9, \beta_2 = 0.999$. The NS-GAN is trained for 200 epochs with a learning rate of $0.0002$. The DiffAugment-GAN is trained with learning rate of $0.001$ for 1 million steps (expect for the case of $4\,000$ training samples, which was extended for 2 millions steps).

## F.7 Performance evaluation

**Test log-likelihood.** To evaluate the reconstruction quality, we use the mean predictive log-likelihood evaluated over the test set. This metric tells us how probable it is that the test targets were generated using the test inputs and our model. Notice that for the case of autoencoder models, the test targets are exactly the test inputs. The predictive likelihood is a proper scoring rule [8] that depends on both the accuracy of predictions and their uncertainty.

For BAE, as done in the literature of Bayesian neural networks (BNNs) [15, 26], we can estimate the predictive likelihood for an unseen datapoint, $\mathbf{y}^*$, as follows

$$\mathbb{E}_{p(\mathbf{w}\,|\,\mathbf{y})}[p(\mathbf{y}^*\,|\,\mathbf{w})] \approx \frac{1}{M}\sum_{i=1}^{M}p(\mathbf{y}^*\,|\,\mathbf{w}_i), \quad \mathbf{w}_i \sim p(\mathbf{w}\,|\,\mathbf{y}),$$

where $\mathbf{w}_i$ is a sample from the posterior $p(\mathbf{w}\,|\,\mathbf{y})$ obtained from the SGHMC sampler.

For VAEs, because the randomness comes from the latent code not the network's parameters, we can use MC approximation to estimate the predictive likelihood as follows

$$\mathbb{E}_{q(\mathbf{z}\,|\,\mathbf{x}^*)}[p(\mathbf{y}^*\,|\,\mathbf{z})] \approx \frac{1}{N}\sum_{i=1}^{N}p(\mathbf{y}^*\,|\,\mathbf{z}_i), \quad \mathbf{z}_i \sim q(\mathbf{z}\,|\,\mathbf{x}^*),$$

where $\mathbf{x}^* \stackrel{\text{def}}{=} \mathbf{y}^*$, and $q(\mathbf{z}\,|\,\mathbf{x}^*)$ is the amortized approximate posterior. In our experiments, we use $N = 200$.

For completeness, we also report the test marginal log-likelihood $p(\mathbf{y})$ of VAEs, which is estimated by the importance weighted sampling (IWAE) method [3]. More specifically,

$$\text{IWAE} = \log\left(\frac{1}{K}\sum_{i=1}^{K}\frac{p(\mathbf{y}^*, \mathbf{z}_i)}{q(\mathbf{z}_i\,|\,\mathbf{x}^*)}\right), \quad \mathbf{z}_i \sim q(\mathbf{z}\,|\,\mathbf{x}^*).$$

It can be shown that IWAE lower bounds $\log p(\mathbf{y}^*)$ and can be arbitrarily close to the target as the number of samples $K$ grows. We use $K = 1000$ in the experiments. The full results of test marginal log-likelihood are reported in Tables 7, 8 and 9.

**FID score.** To assess the quality of the generated images, we employed the widely used Fréchet Inception Distance [12]. This metric is the Fréchet distance between two multivariate Gaussians, the generated samples and real data samples are compared through their distribution statistics:

$$\text{FID} = \|\mu_{\text{real}} - \mu_{\text{gen}}\|^2 + \text{Tr}(\Sigma_{\text{real}} + \Sigma_{\text{gen}} - 2\sqrt{\Sigma_{\text{real}}\Sigma_{\text{gen}}}). \tag{24}$$

Two distribution samples are calculated from the 2048-dimensional activations of pool3 layer of Inception-v3 network [2]. In our experiments, the statistics of generated and real data are computed over 10000 generated images and test data, respectively.

---

[2]We use the original TensorFlow implementation of FID score which is available at https://github.com/bioinf-jku/TTUR.

# G  Additional Results of Comparison with Temperature Scaling

In Bayesian deep learning, *temperature scaling* is a practical technique to improve predictive performance [36, 14, 33]. There are two main approaches to tempering the posterior, namely (1) *partial tempering* and (2) *full tempering* [1, 35]. In this section, we investigate rigorously the posteriors induced by the $\mathcal{N}(0, 1)$ prior and optimized prior under different tempering settings. We use the same setup of MNIST as in the main paper, with 200 examples for inference. For the optimized prior, we use 100 training samples for learning prior. For the $\mathcal{N}(0, 1)$ prior, we use the union of 200 training samples and the data used to optimized prior for training.

## G.1  Partial Tempering

The *partially tempered* posterior is defined as follows [14, 34]

$$p_{\tau_{\text{partial}}}(\mathbf{w} \,|\, \mathbf{y}) \propto \underbrace{p(\mathbf{y} \,|\, \mathbf{w})^{1/\tau}}_{\text{likelihood}} \underbrace{p(\mathbf{w})}_{\text{prior}},$$

where $\tau > 0$ is a *temperature* value. This parameter controls how the prior and likelihood interact in the posterior. When $\tau = 1$ the true posterior is recovered, and as $\tau$ becomes large, the tempered posterior approaches the prior. In the case of small training data and using a misspecified prior such as $\mathcal{N}(0, 1)$, we would use a small temperature value (e.g. $\tau < 1$) to *reduce the effect of the prior*. This corresponds to artificially sharpening the posterior by overcounting the data by a factor of $\tau$.

Fig. 1a shows the test log-likelihood (LL) on MNIST for BAE with $\mathcal{N}(0, 1)$ prior and different temperature values. As expected, the predictive performance of the posterior obtained via low temperatures $\tau < 1$ is much better than those at high temperatures $\tau > 1$. However, cooling the posterior only shows slight improvement compared to the true posterior induced from the optimized prior. In addition, in case $\tau > 1$, where the influence of the posterior becomes stronger, the tempered posterior w.r.t. the optimized prior is significantly better than using the $\mathcal{N}(0, 1)$ prior. This again shows clearly that $\mathcal{N}(0, 1)$ is a poor prior for a deep BAE.

Fig. 2a illustrates samples from priors and posteriors in a low-dimensional space. We also consider the posterior obtained from the *entire* training data and the $\mathcal{N}(0, 1)$ prior as "oracle" posterior. In this case, the choice of the prior does not strongly affect the posterior as this is dominated by the likelihood. It can be seen that, for high-temperature values $\tau > 1$, the *warm posteriors* w.r.t. $\mathcal{N}(0, 1)$ prior are stretched out as the prior effect is too strong. These posteriors are mismatched with the "oracle" posterior as further confirmed by very low test log-likelihood. Meanwhile, due to the good inductive bias from the optimized prior, the corresponding tempered posterior is still located in regions nearby the "oracle" posterior. For low temperature values $\tau < 1$, the *cold posteriors* are more concentrated by overcounting evidence. However, if we use a very small temperature (e.g. $\tau = 10^{-5}$), the resulting posterior overly concentrates around the maximum likelihood estimation (MLE), becoming too constrained by the training data.

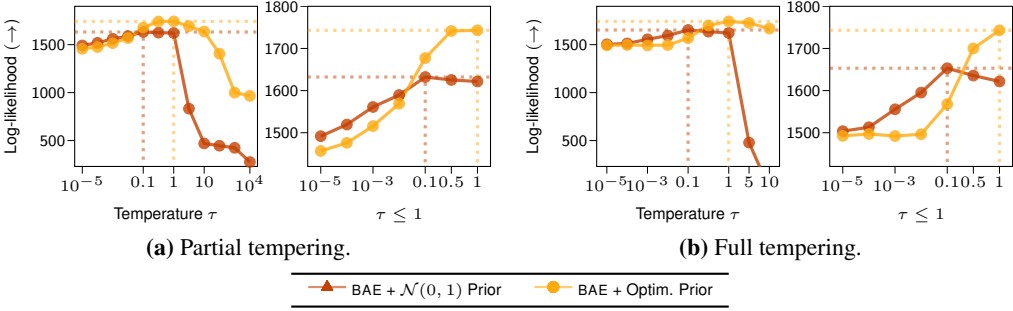

**(a)** Partial tempering.          **(b)** Full tempering.

BAE + $\mathcal{N}(0, 1)$ Prior   BAE + Optim. Prior

**Figure 1:** Test LL as a function of temperature on MNIST using BAE with $\mathcal{N}(0, 1)$ prior. The dotted lines indicate the best performance of LL.

## G.2 Full Tempering

For the fully tempered posterior, instead of scaling the likelihood term only, we scale the whole posterior as follows

$$p_{\tau_{\text{full}}}(\mathbf{w} \mid \mathbf{y}) \propto \big( \underbrace{p(\mathbf{y} \mid \mathbf{w})}_{\text{likelihood}} \underbrace{p(\mathbf{w})}_{\text{prior}} \big)^{1/\tau}.$$

The only difference between partial and full tempering is whether we scale the prior. If we place Gaussian priors on the parameters, this scaling can be absorbed into the prior variance, $\sigma_{\text{full}}^2 = \sigma_{\text{partial}}^2 / \tau$.

Recently, [33] argues that BNNs require a cold posterior, where a $\tau < 1$ is employed, to obtain a good performance. However, we hypothesize that *the cold posterior effect* may originate from using a poor prior. In this case, as shown in Fig. 1b, the results of full tempering are similar to those of partial tempering. Cooling the posterior only helps to increase slightly predictive performance for $\mathcal{N}(0, 1)$ prior. We also observe that the Markov chain Monte Carlo (MCMC) sampling is not converged if a very large $\tau$ is employed, thus we only consider small values of $\tau$ (e.g. $\tau \in \{5, 10\}$). In these cases, as depicted in Fig. 2b, the samples from the posterior may be outside of the hypothesis space of the optimized prior.

In sum, the true posterior induced from our optimized prior is remarkably better than any types of tempered posteriors. These results suggest that, in the small-data regime, we should choose carefully a more sensible prior rather than simply using a vague prior and overcounting the data.

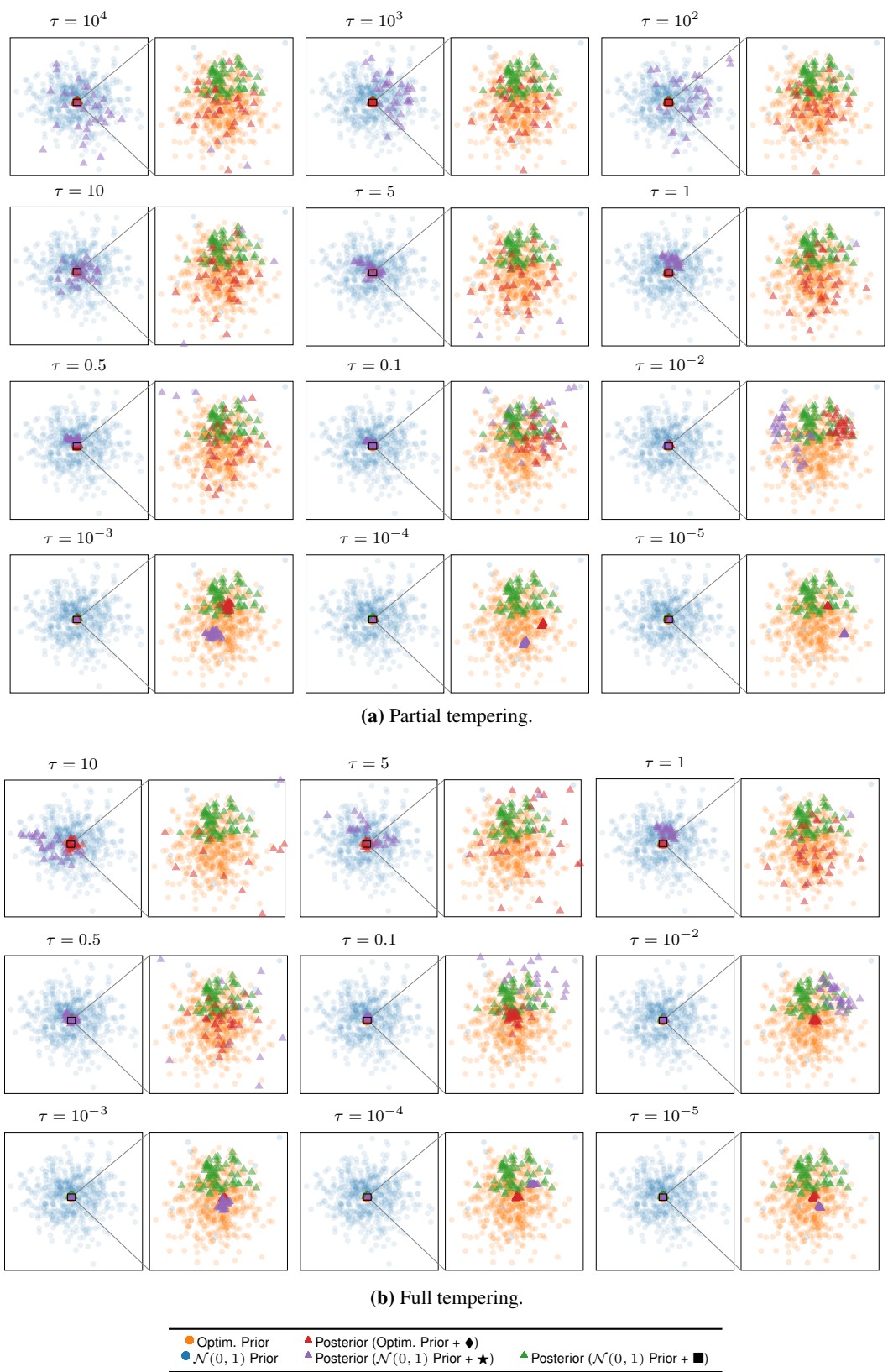

**(a)** Partial tempering.

**(b)** Full tempering.

● Optim. Prior ▲ Posterior (Optim. Prior + ♦)
● $\mathcal{N}(0,1)$ Prior ▲ Posterior ($\mathcal{N}(0,1)$ Prior + ★) ▲ Posterior ($\mathcal{N}(0,1)$ Prior + ■)

**Figure 2:** Visualization of samples from priors and posteriors of BAE's parameters in the plane spanned by eigenvectors of the SGD trajectory. ♦ indicates using 200 samples for training; ★ indicates using the union of these samples and 100 samples used for learning the prior; ■ denotes using all 60000 training samples. Here, $\tau$ is the temperature value used for the ♦ and ★ cases. All plots are produced using convolutional BAE on MNIST.

# H   Ablation Studies

## H.1   Additional results of ablation study on the size of the dataset to optimize priors

In this experiment, we demonstrate that we can obtain a sensible result by using a small number of training instances to optimize the prior. Here, we use a set of 200 samples of 0-9 digits for inference, and another dataset also consisting of 0-9 digits for optimizing the prior. Fig. 5 shows the predictive performance and samples from the posterior. We observe that the performance gain by using more data is not significant. We can achieve sensible results by using only about 10-50 samples for each class. In addition, as illustrated in the low-dimensional space (Fig. 5), the hypothesis space of the prior is not collapsed as we increase the size of the dataset used to optimize the prior. As a result, the predictive posterior is also not concentrated to the MLE solutions as further demonstrated in Fig. 4. This behavior is very different from overcounting the data by using temperature scaling, where the posterior becomes more concentrated as the temperature is decreased. This again demonstrates the practicality of our proposed method in the small-data regime.

## H.2   Effect of the dimensionality of latent space

Fig. 3 illustrates the predictive performance of VAEs and BAEs in terms test LL on MNIST for different size of the latent space and training size. It is clear that BAEs with optimized prior consistently outperforms other competitors across all dimensionalities of the latent space and training sizes.

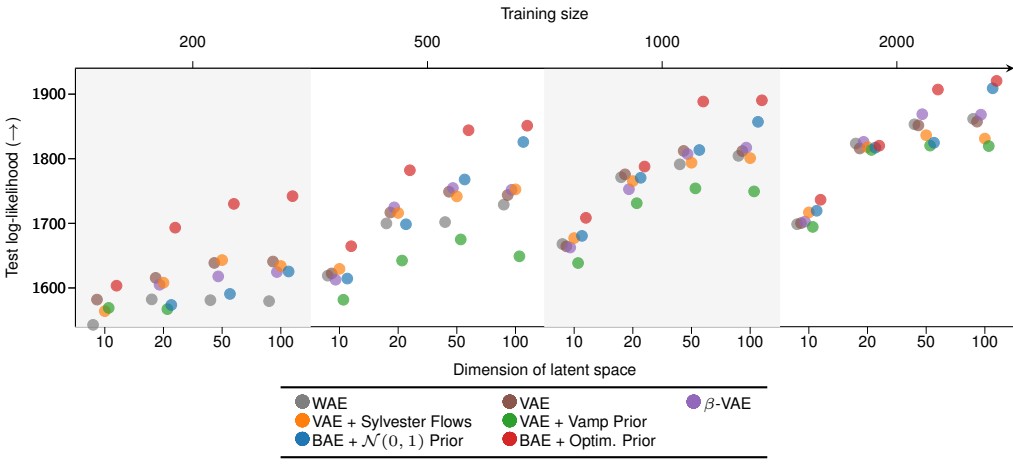

**Figure 3:** Ablation study on the test LL on MNIST dataset for different sizes of the latent space and training sizes.

## H.3   Visualizing 2-dimensional latent space

We run several experiments with a low latent space ($K = 2$) to test the efficacy of VAEs and BAEs as dimensionality reduction techniques. Fig. 6 shows the results, where each color represents an MNIST digit. As seen, BAE with optimized prior produces a more well-defined class structure in comparision with other methods.

We also consider the 2D latent space to visualize that ex-post density estimation with Dirichlet Process Mixture Model (DPMM) helps to reduce the mismatch between the aggregated posterior and the prior. As can be seen from Fig. 7, there are large mismatches between aggregated posterior of VAEs and the $\mathcal{N}(0, 1)$ prior. We can reduce this problem by using a more expressive prior like VampPrior, or performing ex-post density estimation with a second VAE. For BAEs, it is clear that the flexible DPMM estimator effectively fixes the mismatch and this results in better sample quality as reported in the main paper.

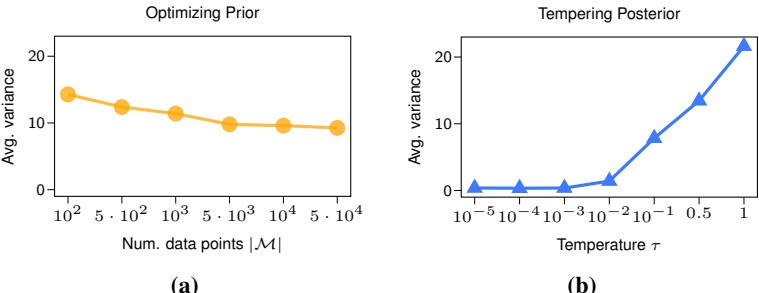

**Figure 4:** The average predictive variance computed over test datapoints as a function of **(a)** the number of data points used to optimize prior, and **(b)** the temperature used for cooling the posterior. Here, we use 200 datapoints from MNIST dataset for inference. In figure **(a)**, we use the optimized prior and consider the true posterior without any tempering. In figure **(b)**, we use the standard Gaussian prior and employ partial tempering for the posterior.

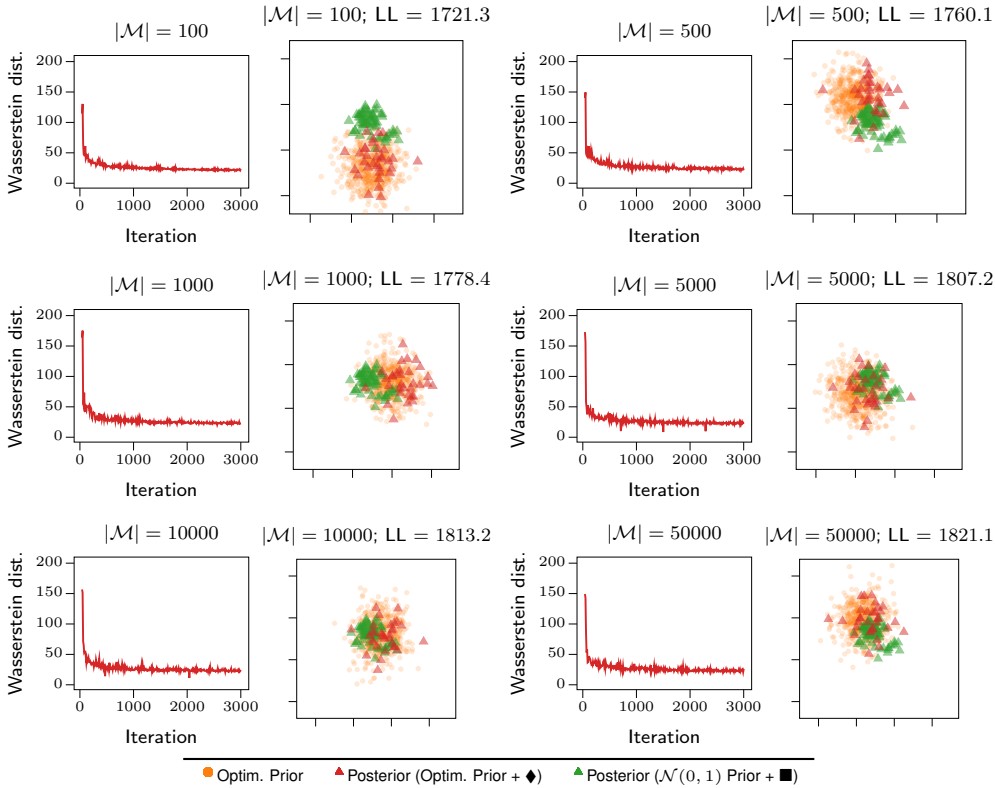

**Figure 5:** Visualization of convergence Wasserstein optimization, and samples from priors and posteriors of BAE's parameters in the plane spanned by eigenvectors of the SGD trajectory corresponding to the first and second largest eigenvalues. Here, $|\mathcal{M}|$ is the size of dataset used for optimizing the prior; ♦ indicates using 200 training samples for inference; ■ denotes using all 60000 training samples for inference; LL denotes the test log-likelihood performance of the posterior w.r.t. the optimized prior. All plots are produced using convolutional BAE on MNIST.

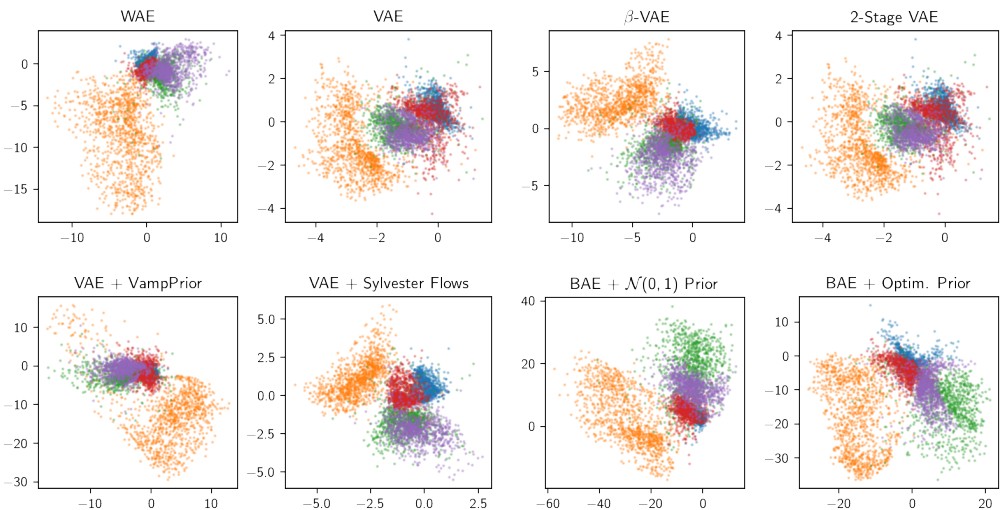

**Figure 6:** Visualization of 2D latent spaces of variants of autoencoders on MNIST test set where each color represents a digit classs. We consider only 5 classes for easier visualization and comparison. All models are trained on 1000 training samples from MNIST dataset.

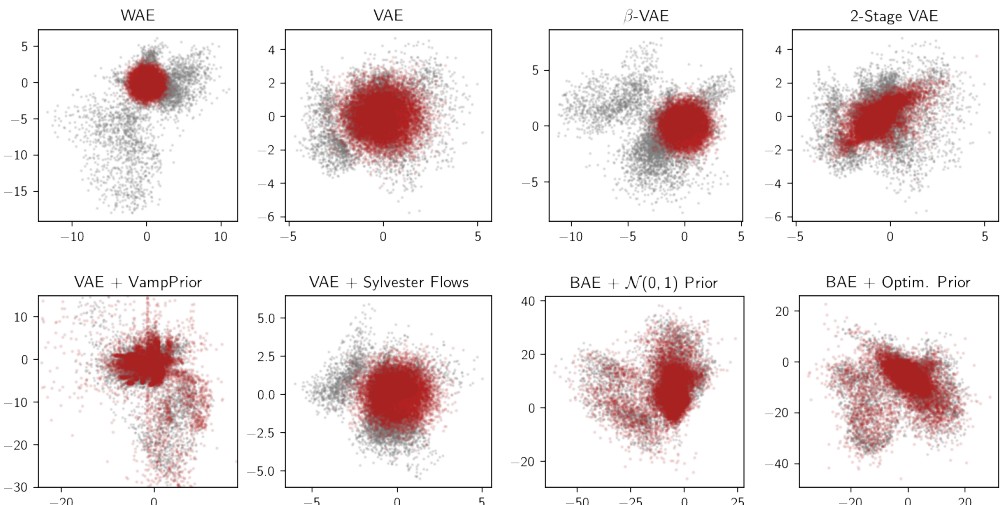

**Figure 7:** Diffrent priors and density estimations on the 2-dimensional latent space of VAEs and BAEs. All models are trained on 1000 training samples from MNIST dataset. The gray points are test set samples while the red ones are samples from priors / density estimators. Here, we employ the isotropic Gaussian prior on the latent space of WAE, VAE, $\beta$-VAE and VAE with Sylveser Flows. The VampPrior is learned to explicitly model the aggregated posterior while 2-Stage VAE uses another VAE to estimate the density of the learned latent space. Meanwhile, for BAEs, we use DPMMs for ex-post density estimation.

# I Additional Results

## I.1 Convergence of Wasserstein optimization

Fig. 8 depicts the progressions of Wasserstein optimization in the MNIST, FREY-YALE and CELEBA experiments.

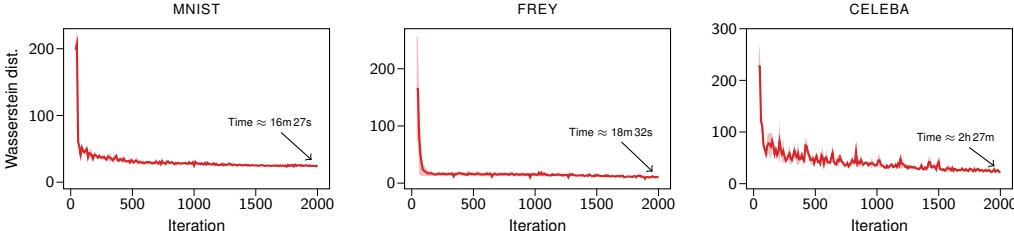

**Figure 8:** Convergence of Wasserstein optimization. The shaded areas represent the standard deviation computed over 4 random data splits.

## I.2 Tabulated results

Detailed results on MNIST, YALE and CELEBA datasets are reported from Table 3 to Table 9.

| | LOG LIKELIHOOD (↑) | | | |
|---|---|---|---|---|
| TRAINING SIZE | 200 | 500 | 1000 | 2000 |
| WAE | 1590.0 (11.0) | 1732.7 (19.2) | 1809.5 (11.1) | 1857.4 (4.8) |
| ★ WAE | 1675.2 (10.6) | 1779.6 (10.2) | 1839.3 (6.1) | 1871.1 (3.1) |
| VAE | 1635.1 (8.0) | 1744.6 (4.5) | 1805.5 (4.8) | 1847.1 (3.7) |
| ★ VAE | 1697.0 (9.9) | 1776.2 (6.8) | 1829.5 (2.8) | 1849.9 (4.4) |
| $\beta$-VAE | 1626.2 (10.3) | 1749.7 (9.2) | 1812.8 (3.8) | 1862.3 (4.9) |
| ★ $\beta$-VAE | 1698.2 (8.0) | 1780.2 (9.3) | 1841.2 (3.4) | 1871.9 (4.4) |
| VAE + SYLVESER FLOWS | 1635.4 (6.1) | 1743.5 (1.5) | 1799.1 (5.5) | 1836.3 (7.2) |
| ★ VAE + SYLVESER FLOWS | 1711.4 (3.0) | 1781.0 (2.9) | 1816.7 (6.2) | 1848.1 (6.5) |
| VAE + VAMPPRIOR | 1543.0 (12.6) | 1669.9 (22.0) | 1756.8 (2.6) | 1818.6 (3.6) |
| ★ VAE + VAMPPRIOR | 1609.6 (14.4) | 1732.1 (14.2) | 1798.1 (5.4) | 1839.3 (4.0) |
| BAE + $\mathcal{N}(0,1)$ PRIOR | 1609.0 (10.6) | 1761.0 (9.1) | 1837.6 (18.4) | 1827.9 (5.7) |
| ★ BAE + $\mathcal{N}(0,1)$ PRIOR | 1681.2 (24.5) | 1798.6 (22.8) | 1827.0 (35.9) | 1842.2 (37.4) |
| BAE + OPTIM. PRIOR (**OURS**) | **1743.5** (12.0) | **1845.1** (1.2) | **1879.1** (6.3) | **1906.8** (1.1) |

**Table 3:** Evaluation of all methods in terms of test log-likelihood (*the higher, the better*) on MNIST. The parentheses are the standard deviations. ★ indicates that we use the union of the training data and the data used to optimize prior to train the model.

| | LOG LIKELIHOOD (↑) | | | |
|---|---|---|---|---|
| TRAINING SIZE | 50 | 100 | 200 | 500 |
| WAE | 689.7 (10.4) | 724.8 (4.4) | 754.5 (3.9) | 787.0 (0.7) |
| ★ WAE | 718.4 (0.9) | 740.6 (4.6) | 765.7 (2.2) | 794.3 (1.6) |
| VAE | 692.3 (8.4) | 723.5 (2.8) | 738.4 (3.2) | 774.1 (1.3) |
| ★ VAE | 701.2 (5.9) | 728.2 (3.5) | 749.4 (2.0) | 774.8 (2.1) |
| $\beta$-VAE | 707.1 (5.7) | 733.8 (8.5) | 761.1 (3.4) | 791.8 (0.7) |
| ★ $\beta$-VAE | 712.1 (7.6) | 737.8 (4.7) | 763.4 (1.3) | 790.8 (1.5) |
| VAE + SYLVESTER FLOWS | 705.4 (4.8) | 729.3 (4.4) | 738.2 (1.6) | 766.8 (0.9) |
| ★ VAE + SYLVESTER FLOWS | 682.1 (11.7) | 716.3 (4.3) | 739.6 (2.1) | 765.3 (1.2) |
| VAE + VAMPPRIOR | 690.0 (6.9) | 722.8 (1.9) | 740.6 (1.8) | 766.8 (2.7) |
| ★ VAE + VAMPPRIOR | 691.7 (6.1) | 716.9 (4.7) | 737.8 (5.3) | 764.2 (2.2) |
| BAE + $\mathcal{N}(0,1)$ PRIOR | 426.1 (27.6) | 668.8 (12.8) | 724.9 (21.2) | 775.5 (4.6) |
| ★ BAE + $\mathcal{N}(0,1)$ PRIOR | 388.0 (13.6) | 570.4 (9.1) | 688.2 (5.1) | 752.5 (1.0) |
| BAE + OPTIM. PRIOR (**OURS**) | **730.3** (3.0) | **754.3** (3.1) | **771.6** (3.0) | **793.5** (2.0) |

**Table 4:** Evaluation of all methods in terms of test log-likelihood (*the higher, the better*) on YALE. The same interpretation as Table 3.

| | LOG LIKELIHOOD ($\uparrow$) | | | |
|---|---|---|---|---|
| TRAINING SIZE | 500 | 1000 | 2000 | 4000 |
| WAE | 5732.6 (35.3) | 6266.4 (73.4) | 6703.6 (24.9) | 6928.3 (32.5) |
| ★ WAE | 6509.7 (49.2) | 6659.8 (30.4) | 6864.0 (23.7) | 7021.6 (24.3) |
| VAE | 5914.2 (78.3) | 6406.4 (39.6) | 6683.6 (87.5) | 6976.4 (11.9) |
| ★ VAE | 6460.1 (33.7) | 6694.1 (63.1) | 6831.8 (97.2) | 7039.5 (36.5) |
| $\beta$-VAE | 5710.2 (49.0) | 6192.5 (91.9) | 6640.6 (139.4) | 7000.9 (7.9) |
| ★ $\beta$-VAE | 6445.3 (94.0) | 6654.6 (44.5) | 6859.0 (39.8) | 7007.7 (86.3) |
| VAE + SYLVESTER FLOWS | 5481.6 (108.4) | 5984.2 (37.4) | 6415.5 (33.5) | 6699.9 (46.9) |
| ★ VAE + SYLVESTER FLOWS | 6241.3 (149.2) | 6437.2 (58.2) | 6519.9 (88.5) | 6831.5 (121.2) |
| VAE + VAMPPRIOR | 5776.6 (95.9) | 6242.2 (92.2) | 6691.5 (24.4) | 6999.7 (15.9) |
| ★ VAE + VAMPPRIOR | 6531.7 (61.5) | 6591.6 (97.4) | 6868.3 (27.8) | 6990.7 (37.3) |
| 2-STAGE VAE | 5914.2 (78.3) | 6406.4 (39.6) | 6683.6 (87.5) | 6976.4 (11.9) |
| ★ 2-STAGE VAE | 6460.1 (33.7) | 6694.1 (63.1) | 6831.8 (97.2) | 7039.5 (36.5) |
| BAE + $\mathcal{N}(0,1)$ PRIOR | 5581.9 (70.8) | 6273.3 (54.2) | 6848.3 (15.1) | 7154.5 (15.6) |
| ★ BAE + $\mathcal{N}(0,1)$ PRIOR | 6574.1 (46.6) | 6826.5 (31.0) | 7038.3 (17.8) | 7223.1 (13.2) |
| BAE + OPTIM. PRIOR (**OURS**) | **6781.3** (32.4) | **7065.8** (15.0) | **7244.7** (8.7) | **7370.0** (13.2) |

**Table 5:** Evaluation of all methods in terms of test log-likelihood (*the higher, the better*) on CELEBA. The same interpretation as Table 3.

| | FID ($\downarrow$) | | | |
|---|---|---|---|---|
| TRAINING SIZE | 500 | 1000 | 2000 | 4000 |
| WAE | 342.14 (19.02) | 309.79 (12.58) | 275.10 (8.71) | 253.06 (5.52) |
| ★ WAE | 294.26 (8.41) | 276.24 (10.49) | 261.64 (6.08) | 246.92 (3.28) |
| VAE | 271.70 (5.12) | 240.69 (3.44) | 230.61 (7.05) | 209.08 (6.28) |
| ★ VAE | 248.18 (12.20) | 237.29 (12.48) | 231.50 (14.17) | 206.92 (9.91) |
| $\beta$-VAE | 323.00 (10.88) | 295.54 (12.45) | 276.71 (15.61) | 250.61 (5.30) |
| ★ $\beta$-VAE | 285.81 (5.58) | 277.44 (12.97) | 271.82 (6.69) | 262.72 (17.92) |
| VAE + SYLVESTER FLOWS | 221.71 (10.50) | 214.94 (12.01) | 207.86 (9.93) | 198.94 (10.10) |
| ★ VAE + SYLVESTER FLOWS | 210.24 (3.48) | 215.00 (5.79) | 204.42 (11.86) | 179.26 (49.53) |
| VAE + VAMPPRIOR | 144.41 (16.61) | 131.02 (2.22) | 112.82 (4.05) | 96.20 (2.79) |
| ★ VAE + VAMPPRIOR | 120.02 (8.62) | 120.23 (7.16) | 102.67 (7.61) | 95.95 (4.86) |
| 2-STAGE VAE | 78.23 (2.56) | 69.37 (2.39) | 67.69 (1.55) | 74.47 (4.52) |
| ★ 2-STAGE VAE | 72.21 (3.05) | 69.25 (3.32) | 72.64 (4.62) | 84.95 (3.91) |
| NS-GAN | 252.33 (27.03) | 171.18 (15.51) | 205.05 (97.46) | 128.29 (3.81) |
| ★ NS-GAN | 151.28 (2.27) | 150.74 (4.39) | 137.64 (4.14) | 139.43 (8.77) |
| ★ DIFFAUGMENT-GAN | **66.09** (0.27) | **58.76** (0.17) | **50.22** (2.62) | **45.14** (0.13) |
| BAE + $\mathcal{N}(0,1)$ PRIOR | 89.36 (4.56) | 81.31 (2.50) | 72.50 (1.37) | 71.85 (0.17) |
| ★ BAE + $\mathcal{N}(0,1)$ PRIOR | 86.03 (3.53) | 75.86 (0.45) | 71.21 (1.41) | 70.72 (0.39) |
| BAE + OPTIM. PRIOR (**OURS**) | 68.59 (3.08) | 66.11 (0.96) | 68.34 (0.86) | 67.18 (0.80) |

**Table 6:** Evaluation of all methods in terms of FID (*the lower, the better*) on CELEBA. The same interpretation as Table 3.

|  | LOG MARGINAL LIKELIHOOD (↑) | | | |
|---|---|---|---|---|
| TRAINING SIZE | 200 | 500 | 1000 | 2000 |
| VAE | 1648.2 (10.1) | 1744.0 (5.6) | 1795.1 (2.6) | 1829.7 (2.5) |
| ★ VAE | 1702.4 (8.9) | 1771.0 (6.7) | 1816.2 (4.6) | 1832.2 (4.8) |
| $\beta$-VAE | 1497.1 (12.5) | 1625.9 (7.7) | 1687.3 (3.8) | 1734.4 (4.2) |
| ★ $\beta$-VAE | 1570.1 (7.8) | 1655.7 (7.9) | 1715.2 (2.9) | 1747.1 (5.6) |
| VAE + SYLVESTER FLOWS | 1627.0 (6.9) | 1709.8 (1.8) | 1755.4 (4.6) | 1783.3 (5.5) |
| ★ VAE + SYLVESTER FLOWS | 1688.0 (3.2) | 1741.8 (2.5) | 1771.4 (3.2) | 1794.8 (5.0) |
| VAE + VAMPPRIOR | 1545.6 (10.5) | 1681.7 (20.2) | 1758.4 (4.1) | 1810.7 (2.2) |
| ★ VAE + VAMPPRIOR | 1616.3 (15.3) | 1737.5 (11.4) | 1795.6 (4.8) | 1829.1 (2.6) |

**Table 7:** Evaluation of all methods in terms of test log marginal likelihood of VAE models (*the higher, the better*) on MNIST. The same interpretation as Table 3.

|  | LOG MARGINAL LIKELIHOOD (↑) | | | |
|---|---|---|---|---|
| TRAINING SIZE | 50 | 100 | 200 | 500 |
| VAE | 693.8 (7.8) | 720.8 (3.4) | 734.5 (2.8) | 767.2 (0.6) |
| ★ VAE | 704.2 (5.6) | 723.7 (3.1) | 742.4 (1.9) | 765.1 (1.2) |
| $\beta$-VAE | 628.1 (2.7) | 655.2 (9.9) | 683.0 (3.9) | 712.5 (1.6) |
| ★ $\beta$-VAE | 658.5 (13.4) | 683.9 (5.4) | 707.2 (2.7) | 731.5 (2.3) |
| VAE + SYLVESTER FLOWS | 668.6 (5.2) | 686.5 (3.4) | 695.1 (1.5) | 718.0 (0.8) |
| ★ VAE + SYLVESTER FLOWS | 655.6 (4.9) | 677.2 (3.8) | 695.7 (0.7) | 717.2 (0.7) |
| VAE + VAMPPRIOR | 672.7 (7.9) | 697.4 (6.8) | 733.4 (3.2) | 759.0 (1.2) |
| ★ VAE + VAMPPRIOR | 703.9 (4.2) | 721.5 (4.0) | 736.8 (4.1) | 760.0 (2.2) |

**Table 8:** Evaluation of all methods in terms of test log marginal likelihood (*the higher, the better*) of VAE models on YALE. The same interpretation as Table 3.

|  | LOG MARGINAL LIKELIHOOD (↑) | | | |
|---|---|---|---|---|
| TRAINING SIZE | 500 | 1000 | 2000 | 4000 |
| VAE | 5973.4 (66.7) | 6416.6 (36.6) | 6673.7 (82.6) | 6943.4 (8.7) |
| ★ VAE | 6470.0 (30.7) | 6676.7 (57.4) | 6807.6 (89.3) | 7001.2 (37.5) |
| $\beta$-VAE | 5496.0 (52.9) | 6007.8 (89.8) | 6457.8 (147.2) | 6820.1 (7.4) |
| ★ $\beta$-VAE | 6294.4 (98.1) | 6472.2 (46.1) | 6680.5 (42.8) | 6844.6 (93.9) |
| VAE + SYLVESTER FLOWS | 5545.8 (97.8) | 5988.2 (40.9) | 6387.9 (37.8) | 6649.9 (47.1) |
| ★ VAE + SYLVESTER FLOWS | 6226.9 (140.3) | 6406.2 (53.7) | 6485.3 (85.4) | 6787.9 (126.9) |
| VAE + VAMPPRIOR | 5842.2 (82.8) | 6273.8 (86.3) | 6682.6 (16.8) | 6984.6 (7.4) |
| ★ VAE + VAMPPRIOR | 6538.3 (62.4) | 6595.9 (93.8) | 6852.3 (18.6) | 6966.1 (28.0) |

**Table 9:** Evaluation of all methods in terms of test log marginal likelihood (*the higher, the better*) of VAE models on CELEBA. The same interpretation as Table 3.

| | LOG LIKELIHOOD ($\uparrow$) | | | |
|---|---|---|---|---|
| TRAINING SIZE | 500 | 1000 | 2000 | 4000 |
| WAE | 7418.8 (123.3) | 8342.0 (73.8) | 8840.3 (26.7) | 9230.2 (0.0) |
| ★ WAE | 8644.3 (72.7) | 8889.5 (58.5) | 9033.6 (97.5) | 9257.9 (68.2) |
| VAE | 7575.3 (60.1) | 8343.9 (42.3) | 8817.6 (124.7) | 9251.8 (23.9) |
| ★ VAE | 8608.2 (15.8) | 8855.1 (65.7) | 9079.7 (50.0) | 9276.7 (12.2) |
| $\beta$-VAE | 7632.0 (115.3) | 8220.7 (161.7) | 8910.7 (33.9) | 9305.2 (14.2) |
| ★ $\beta$-VAE | 8647.3 (30.1) | 8768.6 (60.4) | 9132.0 (22.1) | 9290.8 (87.5) |
| VAE + SYLVESTER FLOWS | 6976.7 (162.5) | 7898.8 (106.1) | 8430.5 (58.6) | 8939.1 (53.5) |
| ★ VAE + SYLVESTER FLOWS | 8240.8 (45.2) | 8446.7 (40.6) | 8700.4 (65.6) | 9045.1 (48.8) |
| VAE + VAMPPRIOR | 7447.5 (77.9) | 8251.2 (39.5) | 8775.4 (40.9) | 9261.8 (14.0) |
| ★ VAE + VAMPPRIOR | 8466.2 (50.7) | 8814.3 (61.2) | 9069.2 (41.1) | 9355.3 (9.5) |
| 2-STAGE VAE | 7575.3 (60.1) | 8343.9 (42.3) | 8817.6 (124.7) | 9251.8 (23.9) |
| ★ 2-STAGE VAE | 8608.2 (15.8) | 8855.1 (65.7) | 9079.7 (50.0) | 9276.7 (12.2) |
| BAE + $\mathcal{N}(0, 1)$ PRIOR | 7097.4 (75.0) | 8299.5 (7.6) | 9009.8 (11.4) | 9326.9 (8.4) |
| ★ BAE + $\mathcal{N}(0, 1)$ PRIOR | 8562.8 (43.8) | 8770.5 (158.6) | 9219.2 (9.2) | 9380.2 (97.2) |
| BAE + OPTIM. PRIOR (**OURS**) | **8975.1 (32.4)** | **9244.3 (15.2)** | **9424.8 (7.2)** | **9629.5 (3.9)** |

**Table 10:** Evaluation of all methods in terms of test log-likelihood (*the higher, the better*) on CELEBA. Here, all models are employ with the *truncated Gaussian likelihood*. The same interpretation as Table 5.

| | FID ($\downarrow$) | | | |
|---|---|---|---|---|
| TRAINING SIZE | 500 | 1000 | 2000 | 4000 |
| WAE | 328.85 (17.16) | 296.43 (7.93) | 291.25 (12.45) | 247.82 (0.00) |
| ★ WAE | 287.16 (22.87) | 273.54 (6.44) | 279.81 (17.56) | 255.96 (13.60) |
| VAE | 299.73 (5.21) | 271.97 (5.50) | 248.95 (13.13) | 235.11 (6.48) |
| ★ VAE | 255.93 (12.86) | 256.20 (4.28) | 242.79 (9.26) | 231.78 (7.74) |
| $\beta$-VAE | 334.00 (9.85) | 322.93 (14.36) | 295.70 (7.74) | 283.72 (4.58) |
| ★ $\beta$-VAE | 307.30 (9.06) | 301.14 (8.67) | 286.14 (7.51) | 276.94 (9.99) |
| VAE + SYLVESTER FLOWS | 238.95 (16.95) | 239.26 (19.51) | 229.78 (8.82) | 217.97 (9.79) |
| ★ VAE + SYLVESTER FLOWS | 231.82 (9.54) | 243.18 (2.56) | 221.53 (5.51) | 206.25 (6.18) |
| VAE + VAMPPRIOR | 127.05 (6.18) | 126.32 (4.19) | 105.52 (3.60) | 97.56 (1.08) |
| ★ VAE + VAMPPRIOR | 110.61 (1.29) | 113.03 (1.67) | 101.26 (3.58) | 88.87 (1.50) |
| 2-STAGE VAE | 97.77 (1.01) | 92.52 (2.81) | 95.63 (3.19) | 101.73 (5.24) |
| ★ 2-STAGE VAE | 90.01 (11.92) | 95.29 (6.39) | 100.32 (2.41) | 105.47 (3.99) |
| BAE + $\mathcal{N}(0, 1)$ PRIOR | 84.11 (4.09) | 72.54 (2.21) | 67.87 (0.61) | 67.00 (0.44) |
| ★ BAE + $\mathcal{N}(0, 1)$ PRIOR | 78.06 (1.42) | 78.13 (6.90) | 66.55 (0.87) | 70.47 (6.95) |
| BAE + OPTIM. PRIOR (**OURS**) | **62.75 (3.61)** | **62.42 (1.20)** | **62.17 (0.89)** | **58.84 (1.26)** |

**Table 11:** Evaluation of all methods in terms of FID (*the lower, the better*) on CELEBA. Here, all models are employed with the *truncated Gaussian likelihood*. The same interpretation as Table 6.

## I.3 More qualitative results

**Figure 9:** Qualitative evaluation for sample quality for autoencoders and GANs on CELEBA. Here, we use 500 samples for training/inference.

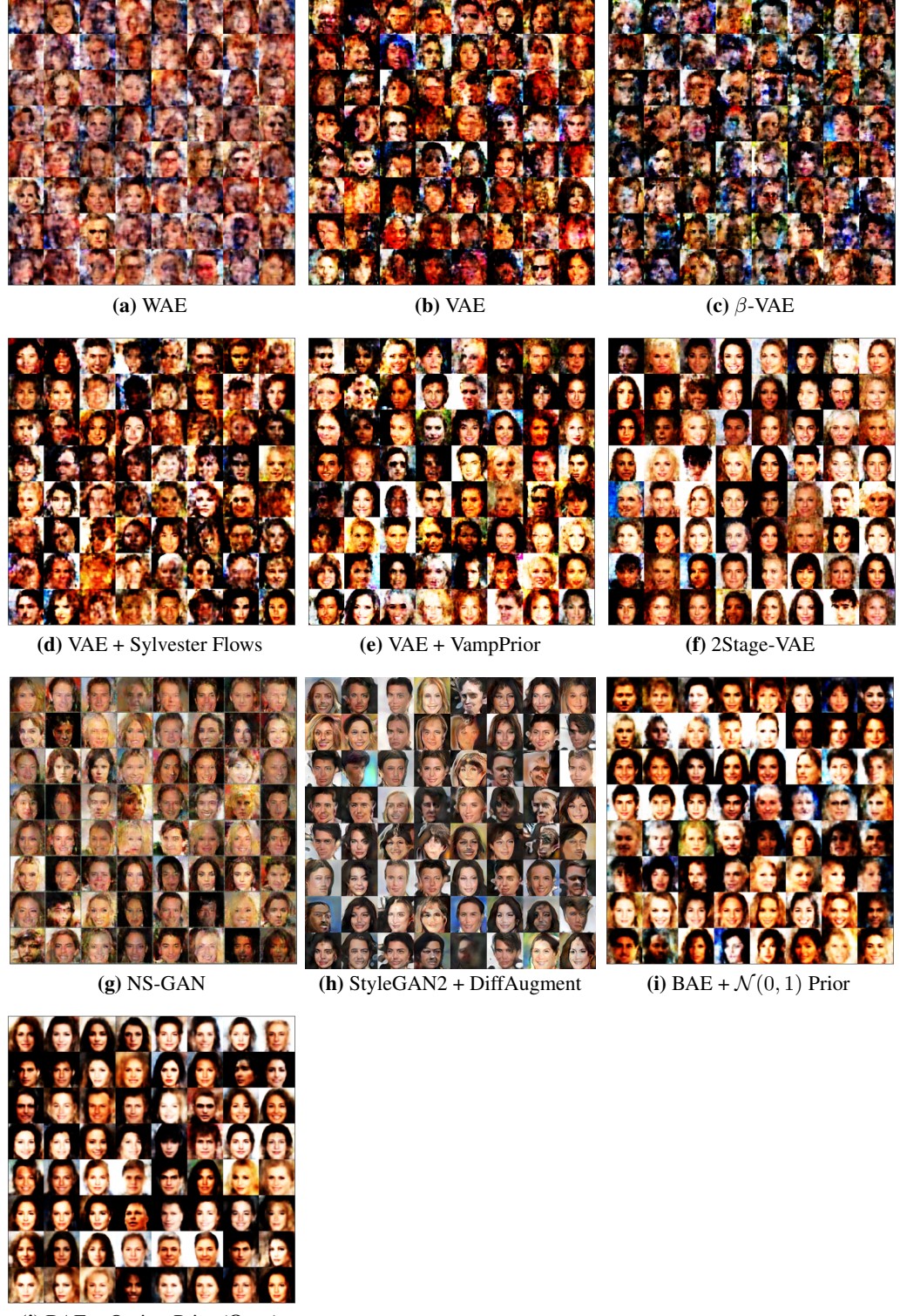

**(a)** WAE       **(b)** VAE       **(c)** $\beta$-VAE

**(d)** VAE + Sylvester Flows       **(e)** VAE + VampPrior       **(f)** 2Stage-VAE

**(g)** NS-GAN       **(h)** StyleGAN2 + DiffAugment       **(i)** BAE + $\mathcal{N}(0,1)$ Prior

**(j)** BAE + Optim. Prior (**Ours**)

**Figure 10:** Qualitative evaluation for sample quality for autoencoders with the *truncated Gaussian likelihood* on CELEBA. Here, we use 500 samples for training/inference.

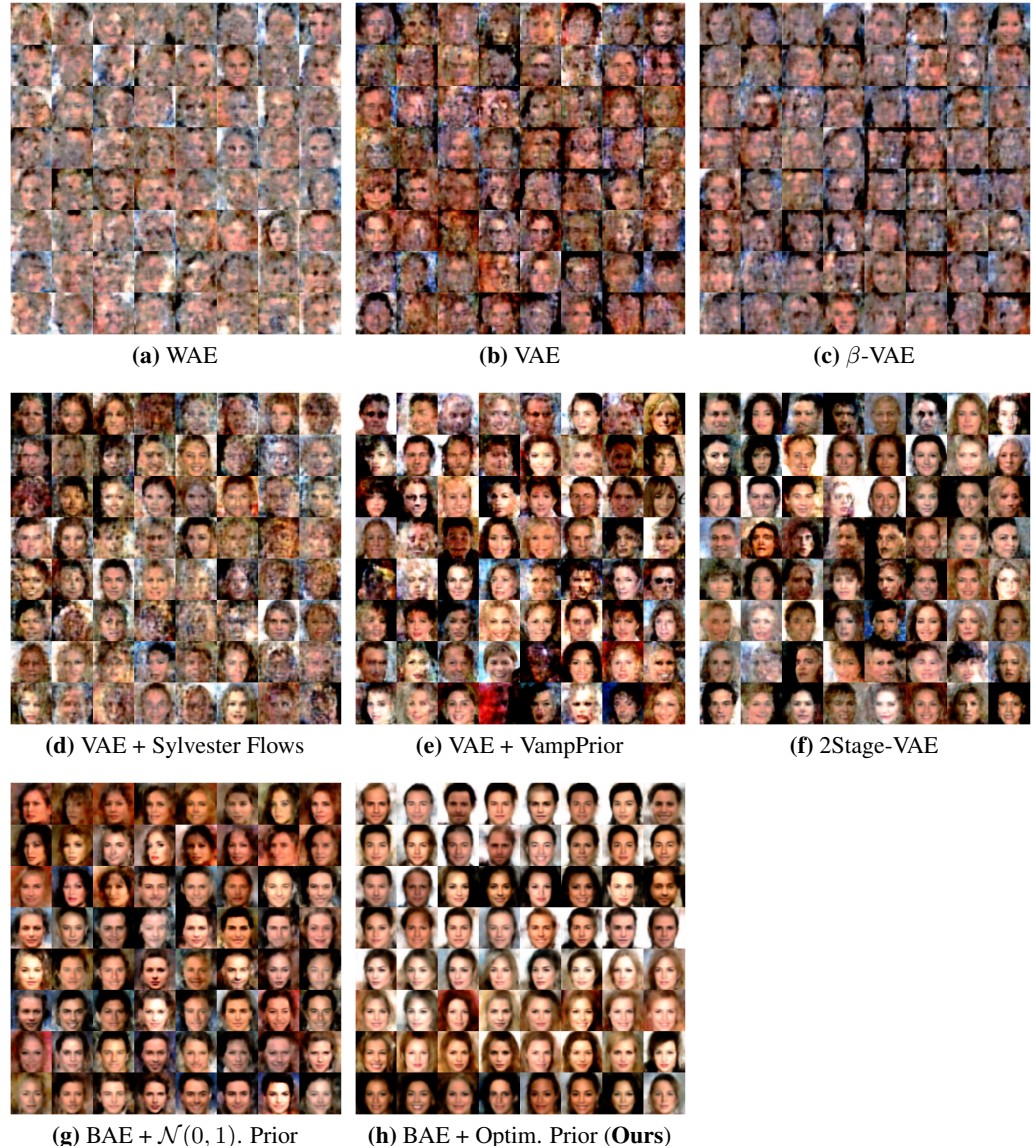

(a) WAE  (b) VAE  (c) $\beta$-VAE

(d) VAE + Sylvester Flows  (e) VAE + VampPrior  (f) 2Stage-VAE

(g) BAE + $\mathcal{N}(0,1)$. Prior  (h) BAE + Optim. Prior (**Ours**)

|  | CELEBA - RECONSTRUCTIONS |
| --- | --- |
| GROUND TRUTH | 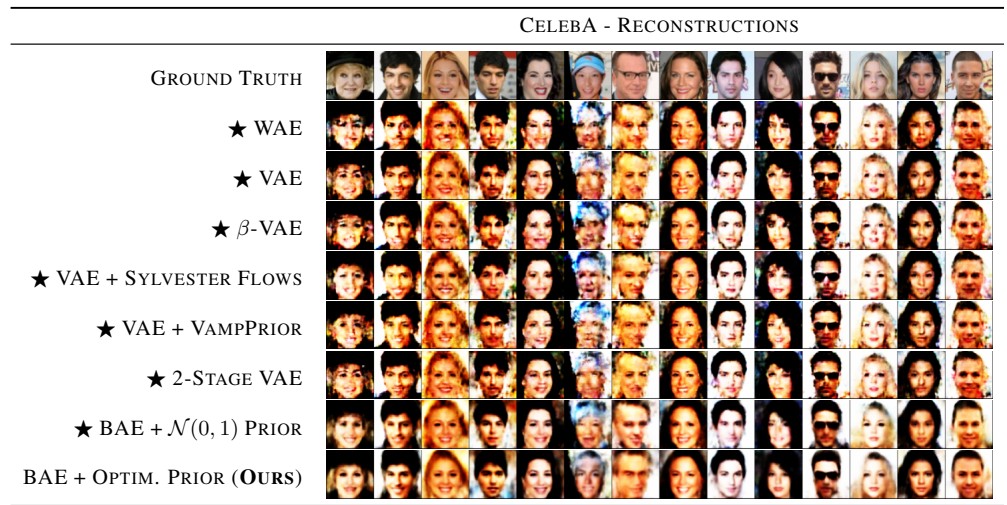 |
| ★ WAE | |
| ★ VAE | |
| ★ $\beta$-VAE | |
| ★ VAE + SYLVESTER FLOWS | |
| ★ VAE + VAMPPRIOR | |
| ★ 2-STAGE VAE | |
| ★ BAE + $\mathcal{N}(0, 1)$ PRIOR | |
| BAE + OPTIM. PRIOR (**OURS**) | |

**Table 12:** Qualitative evaluation for reconstructed samples on CELEBA. ★ indicates that we use the union of the training data and the data used to optimize prior to train the model. Here, the training size is 1000.

|  | CELEBA - RECONSTRUCTIONS |
| --- | --- |
| GROUND TRUTH | 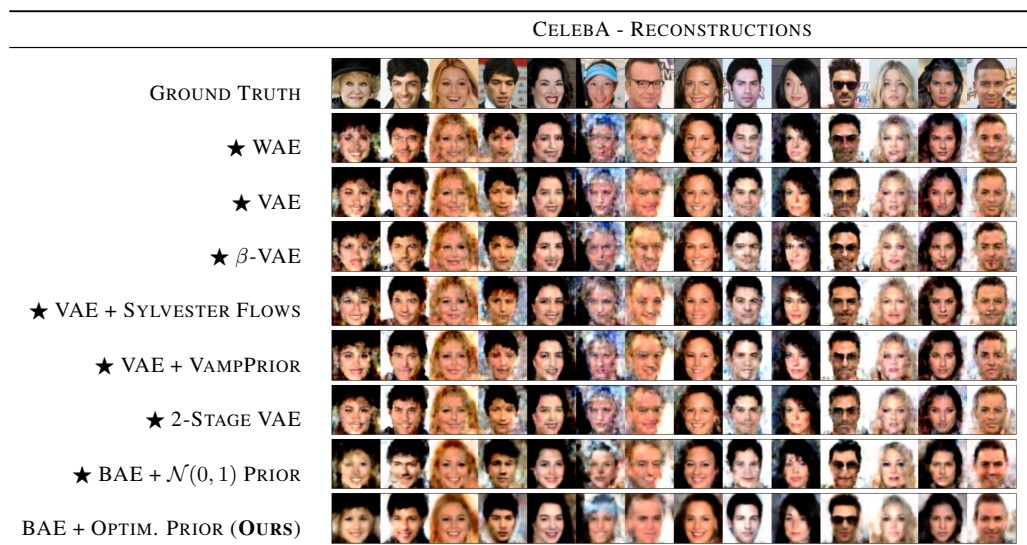 |
| ★ WAE | |
| ★ VAE | |
| ★ $\beta$-VAE | |
| ★ VAE + SYLVESTER FLOWS | |
| ★ VAE + VAMPPRIOR | |
| ★ 2-STAGE VAE | |
| ★ BAE + $\mathcal{N}(0, 1)$ PRIOR | |
| BAE + OPTIM. PRIOR (**OURS**) | |

**Table 13:** Qualitative evaluation for reconstructed samples on CELEBA with the *truncated Gaussian likelihood*. ★ indicates that we use the union of the training data and the data used to optimize prior to train the model. Here, the training size is 1000.

MNIST - RECONSTRUCTIONS

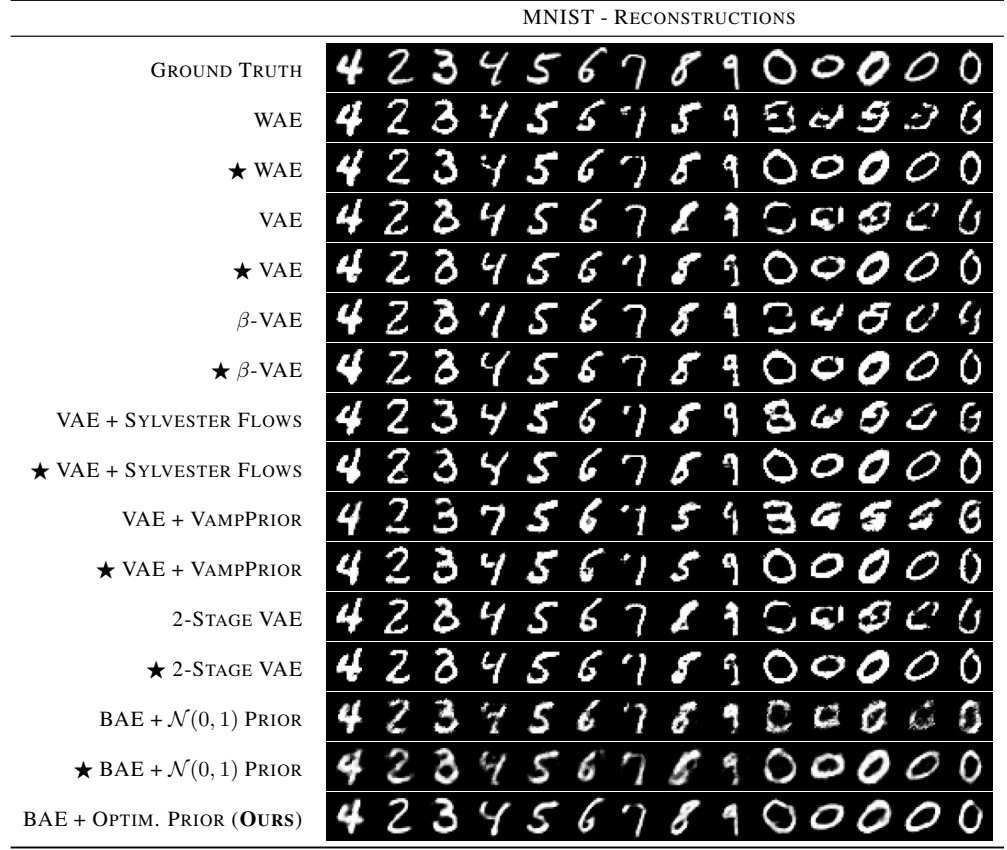

| | |
|---|---|
| GROUND TRUTH | |
| WAE | |
| ★ WAE | |
| VAE | |
| ★ VAE | |
| $\beta$-VAE | |
| ★ $\beta$-VAE | |
| VAE + SYLVESTER FLOWS | |
| ★ VAE + SYLVESTER FLOWS | |
| VAE + VAMPPRIOR | |
| ★ VAE + VAMPPRIOR | |
| 2-STAGE VAE | |
| ★ 2-STAGE VAE | |
| BAE + $\mathcal{N}(0,1)$ PRIOR | |
| ★ BAE + $\mathcal{N}(0,1)$ PRIOR | |
| BAE + OPTIM. PRIOR (**OURS**) | |

**Table 14:** Qualitative evaluation for reconstructed samples on MNIST. ★ indicates that we use the union of the training data and the data used to optimize prior to train the model. Here, the training size is 200.

| MNIST - GENERATED SAMPLES |
|---|

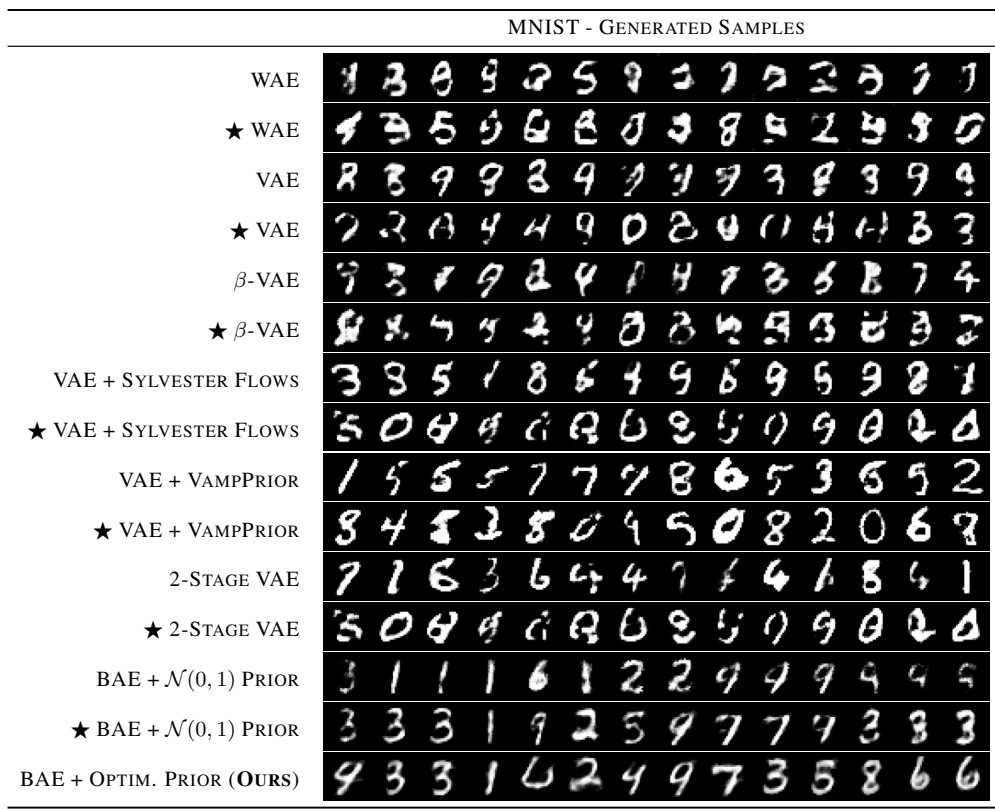

Table 15: Qualitative evaluation for generated samples on MNIST. ★ indicates that we use the union of the training data and the data used to optimize prior to train the model. Here, the training size is 200.

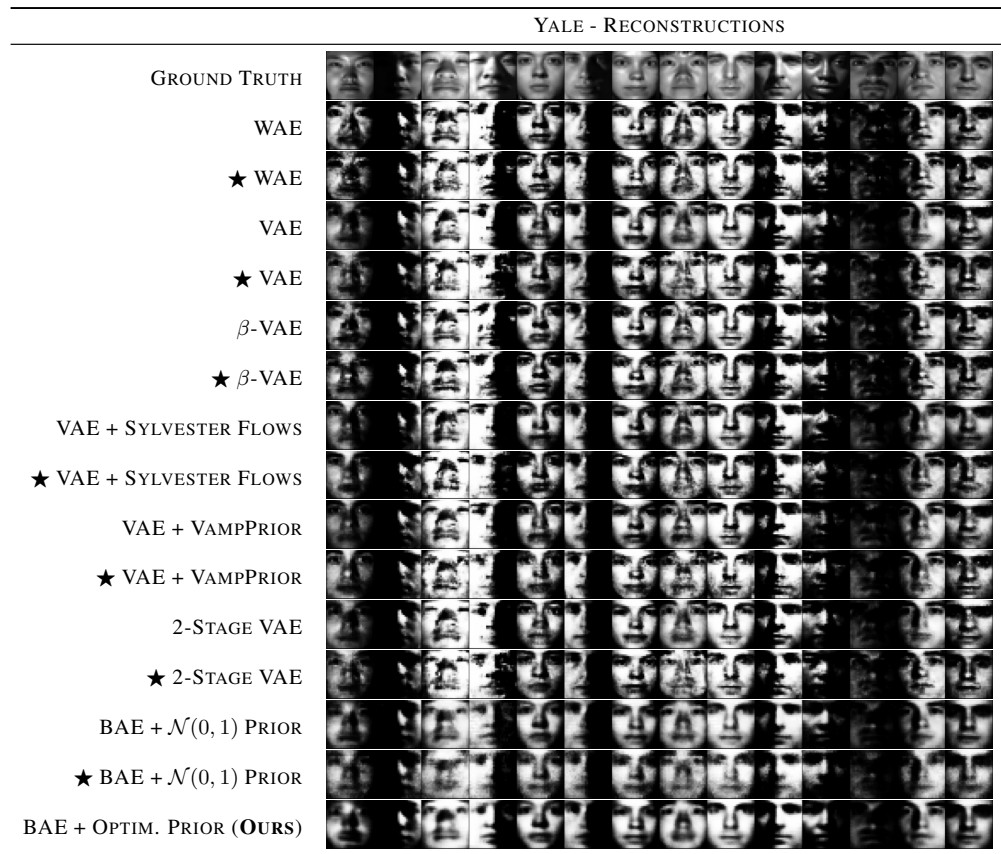

**Table 16:** Qualitative evaluation for reconstructed samples on YALE. ★ indicates that we use the union of the training data and the data used to optimize prior to train the model. Here, the training size is 500.

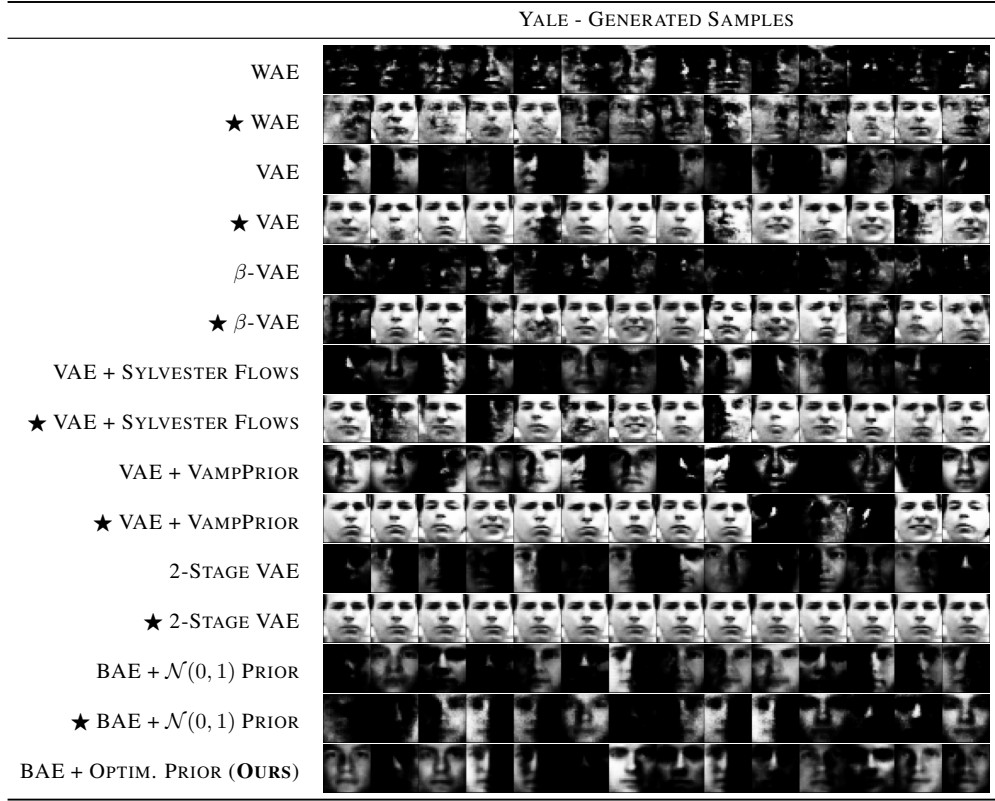

**Table 17:** Qualitative evaluation for generated samples on YALE. ★ indicates that we use the union of the training data and the data used to optimize prior to train the model. Here, the training size is 500.