# OpenReview forum: "Model Selection for Bayesian Autoencoders"
_NeurIPS.cc/2021/Conference — NeurIPS 2021 Poster_

### Official Review · Reviewer_r735 · 2021-07-15

**Rating:** 8
**Confidence:** 5

**Summary:**

The author(s) face the design of Bayesian autoencoders (BAE) and, in particular, the model selection problem. A likelihood model is proposed, then to optimize it the prior of the weights is also needed, a normal distribution is proposed where mean and variances are to be learned. Marginal likelihood estimation is brought here to learn these hyperparameters, but being intractable is transformed into an approximated expression using distributional sliced-Wassersteing distance (DSWD). This expression is estimated with Monte Carlo integration and optimized via gradient descent. In the end, the prior is found by matching the true data distribution and the parametrized marginal distribution of the joint probability of the data and the weights.

The contribution of the manuscript is to develop the mathematical framework to optimize the hyperparameters of a prior for the weights in an autoencoder using DSWD. The solution is designed for a normal prior with to-be-learned mean and variance and continuous Bernoulli likelihood for the probability of the data conditioned to the weights.

**Ethical Concerns:**

There are no ethical concerns, beyond the future application of generative modeling to non-ethical applications.

**Limitations And Societal Impact:**

They have addressed this issue. Being a general approach to learn how to generate samples, it has not immediate societal impact.

**Main Review:**

Although the paper deals with quite involved expressions and methods, it is well written and can be followed. The proposal is explained step by step, with some summaries. Details have been included in the supplementary materials.

The efforts to make the proposal work are remarkable, and results in the experiments are of interest. Convergence times are affordable, see Fig. 3. Results on test log-likelihood, FID and generated samples are reported. In particular, generated samples are of high quality. And results such as Figure 6 in the supplementary material are interesting.

Overall I find the proposal principled, of quality and significant.

**Time Spent Reviewing:**

6

---

> ### Author Response · Authors · 2021-08-09
> **Comments to Reviewer 4 (r735)**
>
> Many thanks for your careful review and detailed feedback!

---

### Official Review · Reviewer_7aYu · 2021-07-16

**Rating:** 4
**Confidence:** 5

**Summary:**

The goal of this paper is to train a Bayesian autoencoder (BAE), where the AE parameters are specified as random variables with trainable hyperparameters. Three contributions are claimed: (1) the distributional sliced-Wasserstein (DSW) distance [25] is leveraged to enable sample-based training of BAE hyperparameters (which model the distribution of AE parameters); (2) to instantiate the AE parameters, the SGHMC [9] is used; (3) to sample from a BAE, the authors propose to learn an ex-post density to model the distribution of the projected latent codes, mimicking [12,8,16]. Experiments on MNIST, FREY-YALE, and CELEBA are conducted.

**Limitations And Societal Impact:**

Yes.

**Main Review:**

Originality. Existing techniques are combined to train BAEs. However, the presented method does not seem to work much better than existing methods. Related work is believed adequately cited.

Quality. The presented method might not be sound.
It's not rigorous/right to define p(x|w) as in Eq (2), because the lambda is encoded with x and there is a cycle within the definition. Similarly for Eq (3); rigorously, this is not a marginal distribution, which means that it might not be right to minimize the DSW distance between pi and p_phi (in Section 3.2).
The experimental results are not strong enough to support it. Besides, the demonstrated results are not convincing and there might be bugs in the code; for example, the reconstructed/generated images show distinct hue characteristics from the training images, which might be caused by the batch-norm layers in the model.

Clarity. Overall, the submission is not in a good shape. It's challenging to figure out what's actually done. Please refer to other comments below.

Significance. The main research direction of Bayesian neural networks is very important. However, the presented method is not convincing.

Other comments:

The word "prior" is confusing; sometimes it means the prior for AE parameters, while sometimes it means the prior for the projected latent codes.

In the Inference paragraph starting from Line 109, it's not clear at all that the inference is to instantiate the AE parameters w from BAE hyperparameters. Similarly for the reasons why we need to do inference.

It's not clear what are the main novelties, as most of the equations in Sections 2 and 3 are from existing papers.

In Lines 200-202, since the sampling is enabled after training (i.e., via the ex-post density estimation of projected latent codes), how do you sample from p_phi (x) for training with DSW distance?

The experimental settings of Section 4.1 are very confusing. I spent a lot of time here, but I failed to understand those settings. For example, what's the meaning of "tuning the prior?"

**Time Spent Reviewing:**

4

---

> ### Author Response · Authors · 2021-08-09
> **Comments to Reviewer 3 (7aYu)**
>
> Thanks for your detailed comments.
>
> - "***The word "prior" is confusing***". There are two main types of autoencoding models considered in this work which are the Variational Autoencoder (VAE) and the Bayesian Autoencoder (BAE). One of the main differences between these two kinds of models is how we impose a prior belief into the model. In particular, we usually place prior on the latent space of VAEs. Meanwhile, for BAEs a prior is imposed on their (weight) parameters. This prior represents our beliefs on the values the BAE parameters should take. This prior on parameters induces a prior distribution over functions/outputs, which is named "functional prior" and defined in Eq. (3) in the main paper. As we mostly care about the distribution over predictive outputs/functions, the prior that matters is the prior in function space. In addition, because of the very large number of BAE parameters involved, it is easier to reason about our preferences for functions rather than parameters.
> - "***Eq (3); rigorously, this is not a marginal distribution***". Following on from the previous comment, Eq. (3) defines the functional prior of BAEs. The marginal distribution is defined by Eq. (5).
> - "***From Line 109, it's not clear at all that the inference is to instantiate the AE parameters w from BAE hyperparameters***". We believe there is a possible misunderstanding here. In contrast to non-Bayesian training of autoencoders (AE), we do not initialize the AE parameters from the BAE optimized prior for later training. Instead, we carry out Bayesian inference for BAEs by means of the stochastic gradient HMC method. In particular, we aim to estimate the *posterior* distribution over BAE parameters by updating our *prior* belief about what values the parameters should take with the *likelihood* of generating the observed data.
> - "***What's the meaning of "tuning the prior?"***". This is one of the main research questions we address in this work. Because BAEs are characterized by a huge number of parameters, it is extremely difficult to choose a sensible prior. Thus, tuning the prior means optimizing its (hyper-)parameters according to the method proposed in this paper. We kindly point the Reviewer to the first bullet of the reply to Reviewer 1 (kcwF), where we further emphasize the motivation of this work. In addition, following on from the previous comment, when the observed data is limited, the posterior is significantly affected by the prior choice. Thus, choosing a good prior is very important in this case. Indeed, our method for optimizing the prior inspired by the empirical Bayesian approach shows significant benefits compared to using vague priors as demonstrated in the experiments.
> - "***The presented method does not seem to work much better than existing methods … The experimental results are not strong enough to support it***". We respectfully but strongly disagree. Figures 2, 4 and 6 in the main paper demonstrate clearly that our proposal works much better than the competing approaches in terms of both qualitative and numerical results. Tabular versions of these figures are presented in Table 3 to 6 in the supplementary material. Furthermore, additional experiments with a completely different likelihood model (a truncated Gaussian) also show that our approach consistently outperforms competing methods (see our response to Reviewer 2 - Lfxg for details).
> - "***It's not rigorous/right to define $p(x|w)$ as in Eq (2), because the $\lambda$ is encoded with $x$ and there is a cycle within the definition***". We agree that the definition of the likelihood in Eq. (2) may raise some confusion. However, it is mathematically correct. Indeed, the BAE is a Bayesian neural network (BNN) used for unsupervised learning problems. For ease of understanding, let's consider BNNs in a supervised learning setting, where each input $x_n$  is associated with a target $y_n$. The likelihood is then $p(y_n | x_n, w) = \text{ContinuosBernoulli}(y_n; f(x_n, w))$; here $\lambda = \hat{y}_n := f(x_n, w)$. For BAEs and autoencoding models in general, the confusion is because of the fact that the target $y_n$ is exactly the input $x_n	$ as we aim to reconstruct the given data $x_n$​​.
> - "***How do you sample from p_phi (x) for training with DSW distance***". It seems that the Reviewer refers to p_psi (x) - the marginal likelihood. If so, in lines 200-202, we already described how to sample from this distribution - “*we can sample from $p_{\psi} (x)$​, by first computing $\hat{x}$​ after sampling from $p_{\psi} (w)$​ and then perturbing the generated $\hat{x}$​ by sampling from the likelihood $p(x | \hat{x})$​*”. In addition, Algorithm 1 in Appendix also describes the procedure to sample from this distribution for training with DSW distance.
> - "***There might be bugs in the code … which might be caused by the batch-norm layers...***". Actually, we did not use batch-norm layers in all experiments as shown in Table 1 in Appendix - network architectures.
> - "***The experimental settings of Section 4.1 are very confusing***". We kindly point the reviewer to the second bullet in the reply to the Reviewer 1 (kcwF), where we emphasize the importance of the choice of data to optimize the prior.
> - "***Overall, the submission is not in a good shape...It's not clear what are the main novelties, as most of the equations in Sections 2 and 3 are from existing papers.***" A summary of the main novelties of this work is presented in lines 49-65. Indeed, the other reviewers agree on the novelty and quality of presentation in this work.

---

> > ### Comment · Reviewer_7aYu · 2021-08-30
> > **Re: Comments to Reviewer 3 (7aYu)**
> >
> > Several concerns of mine were not addressed well.
> >
> > (1) As mentioned in the response, Eq (2) is some kind of reconstruction/conditional distribution, i.e., $x_n \sim p(x_n | \lambda(x_n))$. Similarly Eq (3) also shows a conditional distribution, i.e., $\hat x \sim p(\hat x | x) = \int f(x, w)p(w)dw$. Then with both Eqs (2) and (3), Eq (5) also defines a conditional distribution, which is not a marginal one. Did I misunderstand something?
> >
> > (2) In Figure (6), why are the hue characteristics of the reconstructed/generated images so different from that of the training images, for all AE-kind models?

---

> > > ### Author Response · Authors · 2021-08-30
> > > **Response to Reviewer 3 (7aYu)**
> > >
> > > Thank you for your response.
> > >
> > > - Regarding your 1st concern, we believe there is a possible misunderstanding here.
> > >   - In the paper, we did not claim that Eq. (2) defines a marginal distribution. Instead, we present the likelihood component of the Bayesian model.  Eq. (2) is shown as a conditional distribution because this is how a likelihood model is normally defined. This is presented in Eq. (2) in two equivalent ways:
> > >     - (i) conditional given the model parameters $w$;
> > >     - (ii) conditional given the model output (i.e. the output  $\hat{x}$).
> > >   - Meanwhile, Eq. (3) defines the functional prior obtained by marginalizing out the model parameters $w$, not the input $x$.
> > >   - As we mentioned in lines 142-143 in the main paper, Eq. (5) defines the marginal likelihood obtained by marginalizing out the output $\hat{x}$ and the model parameters $w$​.
> > >
> > > - Regarding your 2nd concern, the hue characteristics of the reconstructed/generated images may be due to the continuous Bernoulli likelihood. We kindly point the Reviewer to our responses to the Reviewer Lfxg, where we discuss about the choice of the likelihood and show new experimental results with a truncated Gaussian likelihood.  The colors in these new results are more appealing. To have fair comparisons, we use the same settings of the network architecture and the choice of the likelihood for all AE-kind models. As demonstrated in the paper and the rebuttal, our method is not only quantitatively but also qualitatively better than the competing approaches in all experiments regardless of using either the continuous Bernoulli or the truncated Gaussian likelihood.
> > >
> > > Once again, thank you for your response. Please feel free to reach out if you have further questions.
> > >
> > > The Authors.

---

### Official Review · Reviewer_Lfxg · 2021-07-16

**Rating:** 8
**Confidence:** 2

**Summary:**

The authors outline 3 limitations of BAEs: 1) lack of generative modeling 2) inference intractability 3) picking good priors over model parameters. They address 1 with latent space density estimation, 2 with MCMC methods, and 3 by replacing the KL between the BAEs induced distribution and the data generating distribution with the distributional sliced-Wasserstein distance.

**Limitations And Societal Impact:**

The author(s) adequately discuss limitations and potential negative societal impact in the last paragraph of their article.

**Main Review:**

### Originality:
- To the best of my knowledge, the proposed objective is novel.

### Quality:
- The continuous Bernoulli is meant to approximate the Bernouli and is appropriate to model (approximately) continuous data in [0,1] with one critical assumption: most of the data is approximately either 0 or 1. Indeed, the continuous Bernoulli can only achieve modes of either 0 or 1. This assumption is true for MNIST, where most pixels are either near 0 or 1. However, using this distribution for grayscale images and/or RGB images where one reasonably expects modes other than 0 and 1 seems drastically inappropriate. Could the author(s) please discuss?
- Why is figure 1 significant? Using a N(0,1) prior on weights amounts to randomly initializing a neural network. What is the expectation here? Ridge regression places N(0,1) on linear coefficients, yet one would never expect anything sensible by sampling coefficients from N(0,1). I think I am missing something. Can the author(s) help me out here?

### Clarity:
- The paper is generally well written.

### Significance:
I am deeply concerned by the choice of likelihood (the continuous Bernoulli). Looking at figure 6's reconstructions for the baseline methods, one can tell pixel intensities for each RGB are close to 0 and 1, which limits the observed colors to red, blue, green, magenta (red + blue), yellow (red + green), cyan (blue + green), white, and black. The proposed method seemingly does a good job of overcoming likelihood misspecification (i.e. interpolating between these 8 color modes). But it must be asked: do the performance benefits hold when the likelihood model is more appropriately selected? If not, I am concerned the complexity of this approach is unwarranted. In other words, does integrating model parameters over a well-selected prior simply provide the proposed methods a way to interpolate between the colors available under the likelihood? If the author(s) could show a tangible benefit for a more appropriate likelihood model, then I would be willing to increase my score significantly.

**Time Spent Reviewing:**

2

---

> ### Author Response · Authors · 2021-08-09
> **Comments to Reviewer 2 (Lfxg)**
>
> Thanks for your insightful and very interesting comment.
>
> - In the Bayesian paradigm, the likelihood represents a modeling choice to be made in addition to the choice of the prior. While we agree with the Reviewer’s point that misspecified likelihoods can severely impact the ability of modeling the data accurately (e.g., Gaussian likelihood for a classification problem), we wanted to remark that since we’re working with image data that are normalized into the range of [0, 1], the continuous Bernoulli likelihood is not such an unreasonable choice (and certainly better than the standard Bernoulli distribution [1]). Indeed, the original paper [1] also considered this likelihood for colored images such as CIFAR-10. Furthermore, in the context of this submission, we stress that our proposed method is not tied to a specific likelihood and that the choice of the continuous Bernoulli was solely out of convenience. However, we agree with the Reviewer that this point is worth a more detailed discussion. During the rebuttal period, we ran an entire new experimental campaign on CelebA, this time using the truncated Gaussian likelihood [2], another valid alternative for [0,1]-valued data which does not bias the pixel values to the extremes. The tables below show the results on CelebA dataset in terms of test log-likelihood and FID score (in parenthesis the std of 3 random seed, for the moment). It’s clear that our proposal still outperforms consistently all the competing methods across all metrics. This further confirms that the benefits from our framework are independent of the choice of likelihood. We thank the Reviewer for raising this concern, which ultimately strengthens even more the message of our paper. For the camera ready, we are including this remark on the choice of the likelihood and these new results in a dedicated subsection of S4.
>
>   | FID Score                     |                  |                  |                  |                  |
>   | ----------------------------- | ---------------- | ---------------- | ---------------- | ---------------- |
>   | ***Training Size***           | ***500***        | ***1000***       | ***2000***       | ***4000***       |
>   | VAE                           | 304.61 (1.71)    | 275.48 (7.80)    | 257.25 (12.69)   | 231.67 (6.55)    |
>   | VAE + VampPrior               | 121.0 (1.9)      | 124.01 (1.34)    | 103.30 (3.93)    | 96.85 (0.81)     |
>   | 2-Stage VAE                   | 96.81 (0.35)     | 94.91 (1.22)     | 93.37 (2.91)     | 98.11 (5.31)     |
>   | BAE + N(0,1) Prior            | 92.82 (4.03)     | 87.15 (3.18)     | 83.11 (0.88)     | 83.64 (0.35)     |
>   | BAE + Optim. Prior (**Ours**) | **75.29 (3.47)** | **71.69 (1.01)** | **78.78 (5.22**) | **76.37 (0.07)** |
>
>
>
>   | Test Log-likelihood               |                   |                  |                       |                  |
>   | ----------------------------- | ----------------- | ---------------- | --------------------- | ---------------- |
>   | ***Training Size***           | ***500***         | ***1000***       | ***2000***            | ***4000***       |
>   | VAE                           | 7614.5 (28.2)     | 8356.1 (35.9)    | 8797.1 (95.4)         | 9263.8 (15.8)    |
>   | VAE + VampPrior               | 7382.4 (31.7)     | 8225.5 (41.6)    | 8795.4 (28.4)         | 9253.2 (10.3)    |
>   | 2-Stage VAE                   | 7614.5 (28.2)     | 8356.1 (35.9)    | 8797.1 (95.4)         | 9263.8 (15.8)    |
>   | BAE + N(0,1) Prior            | 6603.2 (12.4)     | 7842.9 (5.5)     | 8616.7 (17.1)         | 9067.5 (8.9)     |
>   | BAE + Optim. Prior (**Ours**) | **8473.3 (17.9)** | **8949.6 (8.3)** | **9173.5** **(14.2)** | **9378.7 (5.7)** |
>   |                               |                   |                  |                       |                  |
>
> - For the comment on Figure 1, we kindly point the Reviewer to the reply to R1 (kcwF) regarding a similar question.
>
> References:
>
> [1] G. Loaiza-Ganem and J. P. Cunningham. The Continuous Bernoulli: Fixing a Pervasive Error in Variational Autoencoders. NeurIPS, 2019.
>
> [2] Burkardt, J. (2014). The truncated normal distribution. Department of Scientific Computing Website, Florida State University, 1-35.

---

> > ### Comment · Reviewer_Lfxg · 2021-08-17
> > **Post rebuttal reply**
> >
> > Thank you for your reply and the additional results. Your new results are very encouraging. The Continuous Bernoulli may have tested on CIFAR-10, but in my opinion it is best used to model values that are either very near 0 or very near 1. I agree the truncated Normal is much more appropriate. I would even prefer the Normal over the Continuous Bernoulli for RGB images simply because it can attain modes that occur naturally.
> >
> > In my original review, I stated, "Looking at figure 6's reconstructions for the baseline methods, one can tell pixel intensities for each RGB are close to 0 and 1, which limits the observed colors to red, blue, green, magenta (red + blue), yellow (red + green), cyan (blue + green), white, and black." Does the new truncated Normal likelihood improve this? Can you see colors beyond these 8 colors?
> >
> > I will increase my score based on theses new tabular results, but feel the assets (like figure 6) should be generated using this more appropriate likelihood.

---

> > > ### Author Response · Authors · 2021-08-19
> > > **Response to Reviewer 2 (Lfxg)**
> > >
> > > First of all, thank you for your response.
> > >
> > > We were under the impression that we couldn't add links/images in our response, that's why we didn't include it in the previous reply. Since it looks like it's actually allowed, we have generated images with the different models using the truncated Gaussian likelihood ( https://imgur.com/nlCY1L4 - anonymous URL). We agree that the generated images looks a bit more natural and that, as can be seen from the examples, our method is not only quantitatively but also qualitatively better than the competing approaches. For the camera-ready, we're going to remark this discussion: we are indeed updating section 4 to include the previous table and these samples w.r.t. the truncated Gaussian likelihood.
> > >
> > > Once again, thank you for your interesting comment. For additional questions, please feel free to reach out.
> > >
> > > The Authors

---

> > > > ### Comment · Reviewer_Lfxg · 2021-08-19
> > > > **Colors are much more appealing!**
> > > >
> > > > Well done. The color in these images look much much less saturated for all methods, with your method by far looking the best. To clarify is your link for the "reconstruction" or "generate images"? Either way, I highly recommend using these new images in lieu of figure 6.
> > > >
> > > > I further increased my score having been able to observe the new qualitative results.

---

> > > > > ### Author Response · Authors · 2021-08-20
> > > > > **Response to Reviewer 2 (Lfxg)**
> > > > >
> > > > > Regarding your question, the link is for the generated images.  We are glad to see that the rebuttal and this discussion helped to address your remaining concerns.

---

### Official Review · Reviewer_kcwF · 2021-07-21

**Rating:** 6
**Confidence:** 3

**Summary:**

The authors suggest an alternative probabilistic generative model that assumes a prior distribution on the parameters of a classical autoencoder. The authors use empirical Bayes on a separate set of data to determine hyperparameters, use stochastic gradient HMC for posterior inference, and fit a hierarchical latent variable model to the latent codes in order to be able to sample from the latent space.

**Limitations And Societal Impact:**

For limitations see my comments above. I do not think the paper requires a societal impact discussion - at least in its current incarnation.

**Main Review:**

I think this is a well-written paper that constitutes a potentially welcome addition to the increasing arsenal of probabilistic generative models in machine learning. The motivation and presentation is clear, the authors' solution to potential challenges of their methodology is parsimonious, and their results are of interest for other researchers in the field. However, there are also some issues that needs to be resolved before the final decision is made.

Figure 1 summarizes one of my main concerns with the authors' argumentation. From my perspective, this figure demonstrates a result that is completely trivial/expected. Their hyperparameter determination method is an empirical Bayesian approach, so therefore it would of course create a better reconstruction of the data without any training. Actually contrary to authors' presentation, empirical Bayesian approaches are not without their disputes in the field due to their overreliance of the data (the double-dipping problem). So, Figure 1 is not only trivial, but it also represents a potential weakness of the empirical Bayesian methods, which would be overreliance on the data at the expense of generalizability or out-of-distribution performance. Therefore one could expect a sharper decline in reconstruction performance as the data distribution shifts away from the original distribution, or a more drastic loss in local smoothness in the latent space (slight perturbations leading to more drastic changes in the reconstruced output).

The authors' lapse in the presentation of empirical Bayes methods (contrasting them with the 'pathologies' of standard priors) are accompanied by lack of significant experiments that would probe the potential weaknesses of an empirical Bayesian approach. I would love to see how the authors would decide to test these potential weaknesses of their methods. I am not necessarily in favor or against empirical Bayes methods, but if they are used, their potential failure cases must be well examined.

In connection with my previous comments, since there are no theoretical reasoning for setting the prior sample size, I would be very curious as to what kind of results we would get when for authors' method we used all the available data to learn the hyperparameters, and use samples from the 'prior' to conduct inference. Another informative experiment would be to use more uninformative priors such as N(0, 10), so that we can see whether just using more uninformative priors can produce similar results to authors' empirical Bayesian approach.

I also think that the authors' work would benefit a more nuanced discussion of their selection of the DSWD. We do not necessarily know what kind of probabilistic inference this optimization method implicitly corresponds to. Since this is a very challenging issue, this would be fine as long as the authors discuss why DSWD would be preferable over another sample based metric of distributional distance.

**Time Spent Reviewing:**

4

---

> ### Author Response · Authors · 2021-08-09
> **Comments to Reviewer 1 (kcwF)**
>
> Thanks for your insightful comments.
>
> - The results shown in Figure 1 are expected but not trivial, and they serve as motivation for the rest of the paper. First of all, reasoning about functional priors (the prior on the output induced by the marginalization of the prior on the weights) allows us to visualize the pathology of simple standard Gaussian priors. Effectively this choice produces random noise which, as shown in Figure 5, is the result of having an extremely large hypothesis space. At the same time, working with *high-dimensional and well-structured data in addition to over-parameterized models* makes it  extremely difficult to choose a sensible functional prior as these models are characterized by a huge number of parameters.  This is why we need a way to impose a sensible belief to restrict the prior hypothesis space by running a rigorous model selection procedure (as shown in Figure 1 on the right). As suggested by the Reviewer, we have conducted experiments using the more uninformative prior N(0, 10) on MNIST and CelebA datasets. As can be seen from the tables below, the log-likelihood results of this prior degraded considerably compared to those of the N(0, 1) prior in the cases of small training data. In addition, our optimized prior consistently outperforms both these vague priors in all cases. This further confirms the need of an informative prior having a good inductive bias rather than using uninformative priors.
>
>   |  Test Log-likelihood - MNIST           |                   |                  |                  |                  |
>   | ----------------------- | ----------------- | ---------------- | ---------------- | ---------------- |
>   | ***Training size***     | ***200***         | ***500***        | ***1000***       | ***2000***       |
>   | N(0, 1) Prior           | 1609.0 (10.6)     | 1761.0 (9.1)     | 1837.6 (18.4)    | 1827.9 (5.7)     |
>   | N(0, 10) Prior          | 1502.4 (100.7)    | 1656.6 (83.3)    | 1807.2 (34.7)    | 1867.75 (21.1)   |
>   | Optim. Prior (**Ours**) | **1743.5 (12.0)** | **1845.1 (1.2)** | **1879.1 (6.3**) | **1906.8 (1.1)** |
>
>   |  Test Log-likelihood - CelebA          |                   |                   |                  |                   |
>   | ----------------------- | ----------------- | ----------------- | ---------------- | ----------------- |
>   | ***Training size***     | ***500***         | ***1000***        | ***2000***       | ***4000***        |
>   | N(0, 1) Prior           | 5581.9 (70.8)     | 6273.3 (54.2)     | 6848.3 (15.1)    | 7154.5 (15.6)     |
>   | N(0, 10) Prior          | 5415.5 (95.7)     | 6155.2 (8.1)      | 6777.7 (18.5)    | 7118.6 (13.4)     |
>   | Optim. Prior (**Ours**) | **6781.3 (32.4)** | **7065.8 (15.0)** | **7244.7 (8.7)** | **7370.0 (13.2)** |
>
> - We agree with the Reviewer's opinion about possible problems due to the overreliance on the data of the empirical Bayes approach. However, there are many examples where this approach works well in practice, ranging from non-parametric models, e.g. Gaussian Processes [1], to parametric models, e.g. Bayesian Deep Nets [2]. Moreover, in the small-data regime, again, imposing a sensible inductive bias is very important. The Bayesian paradigm allows us to incorporate such a bias into the model via the prior in a principled way. In addition, in our experiments, the inductive bias is also determined by the choice of the dataset used to optimize the prior. It is not necessary to use exactly the same data for both phases of prior optimization and inference. For example, we study rigorously different choices of the data to optimize the prior in the experiment 4.1 as follows
>
>   - *MNIST experiment*: we demonstrate the ability of our approach to incorporate prior knowledge about completely unseen data with different characteristics into the model.
>   - *Frey-Yale experiment*: we demonstrate the benefit of using a different dataset but the same domain e.g. face images.
>
>   In all scenarios, our proposal works much better than the competing approaches despite using less data in the inference phase.
>
> - Regarding the experiments of testing the potential weakness of an empirical Bayesian approach and setting the prior sample size, we already did some comparisons with the temperature scaling approach (Sec. 4.3 and Appendix G) and an ablation study on the size of the dataset to optimize the prior (Sec. 4.3 and Appendix H.1). Temperature scaling is a popular technique in Bayesian deep learning when models are misspecified [3, 4]. This technique is equivalent to replicating the observed data multiple times or applying Bayes’ rule recursively on the same data [5]. Therefore, the resulting posterior w.r.t. using temperature scaling could heavily concentrate on the maximum likelihood estimate, becoming too constrained by the training data (see Figure 2 in Appendix). 	Meanwhile, our proposal behaves completely differently as the posterior after our treatment retains a sufficiently constant variance, regardless of the number of data points used. This means that the hypothesis space of the prior does not collapse as we increase the size of the dataset to optimize the prior. The empirical results in Appendix H.1 also demonstrate that the performance gain of using more data to optimize the prior is not significant. We can obtain a sensible result by using a small amount of data to optimize the prior.
>
> - "***What kind of results ... used all the available data to learn the hyperparameters, and use samples from the 'prior' to conduct inference***". As suggested by the Reviewer, we’ve run this experiment on MNIST. The table below shows the test log-likelihood w.r.t. different numbers of *“pseudo data” sampled from the prior* and used for inference. Indeed, all these results are not bad because of using sensible samples from the prior (as illustrated in Figure 1 in the main paper) for inference. However, these results are much worse than those using the original data (shown in the table above) despite using many more samples for inference.
>
>   | \#Pseudo data | 200          | 500           | 1000         | 2000         | 5000         | 10000         | 20000        |
>   | ------------- | ------------ | ------------- | ------------ | ------------ | ------------ | ------------- | ------------ |
>   | Test log-lik  | 1386.4 (1.9) | 1517.57 (4.8) | 1586.6 (6.8) | 1608.2 (9.7) | 1625.2 (9.5) | 1651.8 (23.6) | 1696.5 (1.0) |
>
> - Regarding the choice of the objective to optimize, in order to be practically feasible the proposed method had to address two major constraints: computational tractability and scalability to high-dimensional spaces. Both of these are very efficiently tackled using the DSWD formulation of the Wasserstein distance. We will emphasize this in the camera-ready version of the paper.
>
>   References:
>
>   [1] C. E. Rasmussen and C. Williams. Gaussian Processes for Machine Learning. MIT Press, 2006.
>
>   [2] Alexander Immer et al. Scalable Marginal Likelihood Estimation for Model Selection in Deep Learning. ICML, 2021.
>
>   [3] L. Aitchison. A Statistical Theory of Cold Posteriors in Deep Neural Networks. ICLR, 2021.
>
>   [4] A. G. Wilson and P. Izmailov. Bayesian Deep Learning and a Probabilistic Perspective of Generalization. NeurIPS, 2020.
>
>   [5] Subhash R. Lele et al. Estimability and Likelihood Inference for Generalized Linear Mixed Models Using Data Cloning. Journal of the American Statistical Association Vol. 105, No. 492 , 2010.

---

> > ### Comment · Reviewer_kcwF · 2021-08-30
> > **Thank you for the response**
> >
> > I thank the authors for their response and the effort they put in resolving my concerns. I raise my score to reflect my improved opinion of their work.
> >
> > I would encourage the authors to include a discussion as to the potential dangers of empirical Bayesian methods, accompanied by a discussion of why and when they work well, and why in this in their opinion it works well. I still dispute the uncritical presentation of the empirical Bayesian methods, and hope that the authors will opt for a more nuanced discussion.
> >
> > "Regarding the experiments of testing the potential weakness of an empirical Bayesian approach..."  Perhaps the effects of distribution shift could be examined as a potential failure case for the authors' approach? This would be a good stress test for a method that invokes data-based priors.

---

> > > ### Author Response · Authors · 2021-08-31
> > > **Thank you for your response**
> > >
> > > Thank you for your insightful feedback. For the camera-ready, we are going to include a discussion of the potential issues of the empirical Bayes approach.

---

### Author Response · Authors · 2021-08-09
**General comments to Reviewers**

First of all, we would like to thank the Reviewers for their comments and helpful feedback.

With this paper we analyze a Bayesian treatment of auto-encoders and we identify the difficulty of imposing sensible priors on the model parameters as one of the main reasons why BAEs aren’t as popular as other generative models. We move then to propose a methodology to restrict the hypothesis space defined by the choice of prior using recent advances on scalable Wasserstein distance computation, and this is done by drawing an important connection with the empirical Bayes method, a simple, powerful and reliable way to do Bayesian model selection.

We are delighted to see that Reviewers agree on the novelty and quality of presentation (R1 (kcwF), R2 (Lfxg), R4 (r735)), that our proposal is principled, of quality and significant (R4) with results that are of interest for other researchers in the field (R1).

In the threads below we will address the Reviewers’ comments and questions in detail.

---

### Decision · Program_Chairs · 2021-09-27

**Decision:**

Accept (Poster)

**Comment:**

This paper introduces a model selection method for Bayesian autoencoders, based on a Bayesian formulation. Learning is achieved by optimization of the distributional sliced-Wasserstein distance (DSWD) and posterior estimation is carried out via stochastic gradient Hamiltonian Monte Carlo, useful for ​representation learning with uncertainty.

The paper has been perceived quite positively. The address challenges in working with BAE 1) lack of generative modeling 2) inference intractability 3) picking a prior over model parameters. They address 1 with latent space density estimation, 2 with MCMC methods, and 3 by optimizing the distributional sliced-Wasserstein distance of BAEs induced distribution and the data generating distribution. The authors also provide solid and convincing experimental results.

However, there is also a potential misleading problem with the notation, that is related to the issue raised by Reviewer 7aYu, the apparent cyclic definition in the probabilistic model. In this respect, it is important to highlight that an autoencoder (or its Bayesian treatment BAE) should be viewed as a supervised models This is in contrast to a VAE where the encoder is an inference distribution so strictly speaking, the statistical model is the decoder only.

To highlight the supervised nature of the problem, one remedy could be to introduce $y$ as a supervised target
and let $y_n = x_n$.  Eq (1) can be expressed in the familiar form as in regular Bayesian NNs:
$p(w|x,y) = p(y|w,x) p(w) / p(y|x)$.

Starting with this notation seems to solve the potential confusion (raised as a concern by reviewer 7aYu) that I feel is not fully resolved during the discussion and must be addressed in the final manuscript. I suggest that the authors do not hide the notation and clearly define the Bayesian hierarchical model, and explicitly spell out the conditional in (2).
It could be also helpful to shift the discussion in the experimental section, that starts from line 214 about the key differences between a VAE to the introduction.

I believe that the authors can address this in the final manuscript, hence I am suggesting acceptance as a poster, acknowledging positive points raised by the reviewers.